# Trajectory Balance with Asynchrony: Decoupling Exploration and Learning for Fast, Scalable LLM Post-Training

**Brian Bartoldson**[1]    **Siddarth Venkatraman**[2,3]    **James Diffenderfer**[1]    **Moksh Jain**[2,3]
**Tal Ben-Nun**[1]    **Seanie Lee**[4]    **Minsu Kim**[2,4]    **Johan Obando-Ceron**[2,3]
**Yoshua Bengio**[2,3,5]    **Bhavya Kailkhura**[1]

[1]Lawrence Livermore National Laboratory, [2]Mila – Quebec AI Institute
[3]Université de Montréal [4]KAIST [5]CIFAR Fellow
{bartoldson,diffenderfer2,kailkhura1}@llnl.gov
{siddarth.venkatraman,moksh.jain}@mila.quebec

## Abstract

Reinforcement learning (RL) is a critical component of large language model (LLM) post-training. However, on-policy algorithms used for post-training are not naturally robust to a diversified content of experience replay buffers, which asynchronous off-policy actors can efficiently populate in parallel to training. We propose efficiently learning on such off-policy data via Trajectory Balance with Asynchrony (TBA), an approach to asynchronous RL for LLMs that leverages the principled off-policy TB objective. On math, preference-tuning, and automated red-teaming tasks, we post-train models ranging from Pythia 410M to Qwen 2.5 7B, finding TBA offers speed and performance boosts over strong baselines like Online DPO and Dr. GRPO. Beyond TBA's performance benefits (high accuracy even as asynchrony grows) and speedups ($4\times$ or more), we show its reward- and recency-prioritizing sampling enable further gains as data generation is scaled. Our code is available at https://github.com/bbartoldson/TBA.

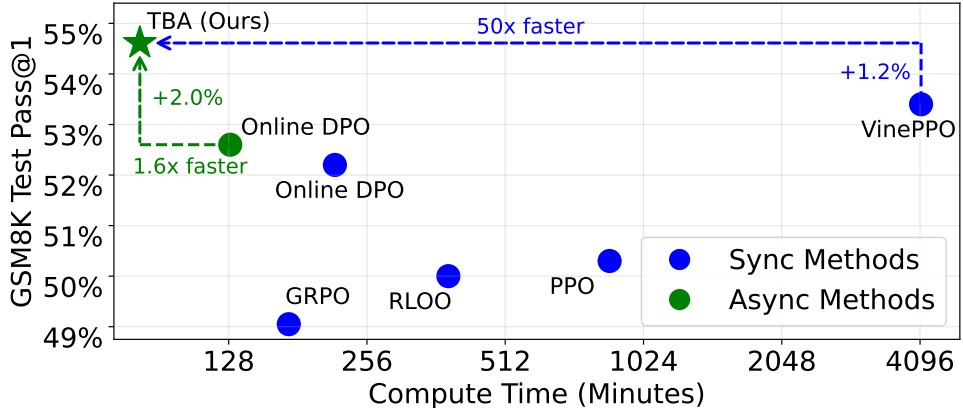

Figure 1: **TBA excels on the GSM8K mathematical reasoning task**. All plotted points use 4xA100 GPUs (or comparable L40S GPUs). DPO and RLOO baselines taken from Noukhovitch et al. [46], PPO and VinePPO baselines taken from Kazemnejad et al. [29]. The baseline model is the SFTed RhoMath-1B [35] model, which gets 40% accuracy after SFT and before RL. Appendix B has details.

39th Conference on Neural Information Processing Systems (NeurIPS 2025).

# 1 Introduction

Post-training through reinforcement learning (RL) is critical for enhancing large language models (LLMs), aligning them with human preferences and improving their reasoning abilities [8, 27]. However, widely used RL algorithms such as Proximal Policy Optimization (PPO) [57] and REINFORCE Leave-One-Out (RLOO) [2] suffer from a fundamental limitation: they are *on-policy*, meaning that data generation and policy updates occur *sequentially*. This dependence creates *bottlenecks* that reduce resource utilization. Further, benefits of scaling on-policy data generation may be limited [22].

We introduce **Trajectory Balance with Asynchrony (TBA)**, an asynchronous RL approach designed to efficiently and scalably leverage compute for LLM post-training. TBA uses an *off-policy training objective* based on Trajectory Balance (TB) [39], which we hypothesize can improve learning when using async RL to decouple data generation from policy updates, mitigating key RL bottlenecks.

Across *mathematical reasoning*, *preference-tuning*, and *automated red-teaming* tasks, we find TBA broadly offers three advantages over existing LLM post-training approaches: 1) **Stable off-policy RL**, unlocking asynchrony for massive parallelization and reduced wall-clock times – see Figures 1 and 3. 2) **Sampling from a diverse replay buffer**, improving exploration and preventing mode collapse. 3) **Scalable search compatibility** that aids *sparse reward settings* like automated red-teaming.

Our key contributions are summarized as follows:

- We introduce **TBA**, a distributed asynchronous RL framework for fast, scalable LLM post-training.
- We show that **LLMs post-trained with TBA match or exceed performances from existing methods**, illustrating TB's ability to use off-policy data in asynchronous RL.
- We **demonstrate significant speedups** ($4\times$ to $50\times$) for RL across mathematical reasoning, preference-tuning, and automated red-teaming.

By enabling high-quality and fast off-policy post-training, TBA contributes to *scalable and effective LLM alignment*, ensuring that large models can be refined more efficiently for real-world deployment.

# 2 Related Work

**RL fine-tuning of language models**    Reinforcement learning (RL) has been an integral component for training LLMs [14, 48]. In particular, RL has become the *de facto* approach for aligning language models with human preferences [8, 82, 62, 49]. Much of this work relies on Proximal Policy Optimization (PPO) [57], an on-policy RL algorithm that has become a default choice for fine-tuning LLMs due to its strong performance across different setups. Aside from PPO, other on-policy objectives such as REINFORCE [2] and variants like GRPO [59] and VinePPO [29] have also been studied in the context of language models. An alternative to PPO-based fine-tuning is rejection sampling fine-tuning, inspired by the best-of-$n$ inference approach proposed by Nakano et al. [45]. Recent work by Dong et al. [10], Gulcehre et al. [15], and Wang et al. [72] extends this concept by generating $n$ candidate responses for each prompt, ranking them with a learned reward function, and fine-tuning the model based on the highest-ranking responses. Separately, direct preference learning approaches [53, 3, 64] skip reward modeling entirely and train language models to directly optimize responses under a preference model. Hu et al. [23] introduced GFlowNet fine-tuning, leveraging off-policy GFlowNet algorithms for fine-tuning language models, which we build upon here.

**Asynchronous distributed RL**    Distributed RL spreads actors/searchers, learners, and environments across a collection of computing resources. Asynchronous distributed RL does this such that searcher and trainer processes do not necessarily share the same weights, which can significantly improve training speed [44, 73] and facilitate RL in complex, high-dimensional domains [19, 24, 21].

A foundational method in this area is Asynchronous Advantage Actor-Critic (A3C) [42]. In A3C, multiple parallel workers asynchronously interact with the environment and communicate gradients to a central node. Our approach to async distributed RL more closely resembles the Importance-Weighted Actor-Learner Architecture (IMPALA) method [11], which communicates experience trajectories (state, action, and reward tuples) to the central node. As opposed to standard RL environments, where value-based off-policy learning objectives enable principled async training, language models present a distinct challenge since learning value functions is challenging [16]. Recent work has also leveraged these Async RL frameworks within the context of LLM post-training for training on *near* on-policy data [46, 13]. In contrast to these recent works, our approach uses a principled off-policy learning objective, allowing training on any off-policy trajectories.

**Automated red-teaming** Through adversarial interactions, LLM red-teaming clarifies the robustness and risks of a target LLM. Automating the generation of these adversarial scenarios, automated red-teaming frameworks can help uncover vulnerabilities, biases, and unintended behaviors in models more quickly, enabling preemptive mitigation before deployment [74, 28].

Perez et al. [51] proposed training language models using RL to discover prompts that elicit harmful responses from some target LLM. Standard RL approaches, however, are susceptible to mode collapse and fail to achieve the attack diversity that is critical for successful red-teaming. Hong et al. [20] introduced a curiosity bonus to encourage generation of diverse red-teaming prompts, whereas Samvelyan et al. [56] proposed sampling an attack prompt from a pool and iteratively mutating the prompt with auxiliary LLMs. Lee et al. [34] proposed using GFlowNet fine-tuning followed by MLE smoothing to generate diverse, transferable, and effective prompts. Our red-teaming experiments augment their TB objective optimization with our distributed asynchronous framework.

## 3 Preliminaries

### 3.1 KL regularized RL as probabilistic inference

We study the problem of fine-tuning a pretrained language model $\pi_{\text{ref}}$ with a reward model $r_\phi$. For mathematical reasoning [9] our reward model simply computes a response's correctness, while our preference-tuning for alignment [82] and red teaming [51] experiments use reward models optimized with human preference data. Notably, reward maximizing RL with learned reward functions is susceptible to spurious modes of the reward [61, 50, 12], which can result in poor performance and low diversity in responses. This is addressed by constraining the fine-tuned model to be close to the initial model in terms of the KL divergence:

$$\pi^* = \arg\max_\pi \mathbb{E}_{\mathbf{x} \sim \mathcal{D}}[\mathbb{E}_{\mathbf{y} \sim \pi(\mathbf{y}|\mathbf{x})}[r_\phi(\mathbf{y}; \mathbf{x})] - \beta \mathbb{D}_{\text{KL}}(\pi(. \mid \mathbf{x}) || \pi_{\text{ref}}(. \mid \mathbf{x}))]. \tag{1}$$

Online RL algorithms such as PPO [57, 62] and REINFORCE [75, 31, 2] can be used to optimize the model (which is the policy), whereas offline objectives such as DPO [53] can be used to train the model directly on the preference data and achieve the same optimal policy asymptotically.

Eq. 1 can be interpreted from a probabilistic perspective as a Bayesian posterior inference problem [32]. The optimal policy for Eq. 1 is given as:

$$\pi^*(\mathbf{y} \mid \mathbf{x}) \propto \pi_{\text{ref}}(\mathbf{y} \mid \mathbf{x}) \exp(\beta^{-1} r_\phi(\mathbf{y}; \mathbf{x})). \tag{2}$$

Approaches such as Gibbs sampling can be used to produce samples from the optimal policy without any fine-tuning [76]. These MCMC approaches can be very expensive at inference time and intractable for long sequences. Within the probabilistic interpretation, on-policy RL to maximize Eq. 1 is equivalent to amortized variational inference to minimize the reverse KL with respect to the posterior density [32]. However, reverse KL optimization is susceptible to mode collapse and requires on-policy samples. The reliance on on-policy samples can limit the scalability as we cannot use replay buffers that can be populated in parallel at scale, since new updates of the policy invalidate the on-policy nature of the older replay buffer examples. In practice, this means that the policy can get stuck in suboptimal solutions and stop exploring, or may obtain high reward at the cost of diversity. This motivates us to consider an alternative off-policy amortized variational inference to efficiently leverage scalable computational resources for flexible exploration.

### 3.2 GFlowNets

Generative Flow Networks [GFlowNets; 4, 5] are a framework for off-policy training of hierarchical generative models to sample proportionally to a given unnormalized density (reward) function. GFlowNets frame probabilistic inference as a sequential decision-making problem, learning a policy to construct the objects (e.g. sequences) by putting together building blocks (e.g. tokens) and optimizing consistency-based objectives. GFlowNet objectives have been used for fine-tuning autoregressive [23, 34] as well as discrete diffusion language models [68]. GFlowNets are MaxEnt RL algorithms [81], and notably equivalent to path consistency learning [43] in sequence generation problems [4].

To fine-tune a language model to sample from Eq. 2, we can set as a reward $R(\mathbf{y}; \mathbf{x}) = \pi_{\text{ref}}(\mathbf{y} \mid \mathbf{x}) \exp(\beta^{-1} r_\phi(\mathbf{y}; \mathbf{x}))$. Following Lee et al. [34], we use the *trajectory balance* objective [40] for training the language model policy $\pi_\theta$, which is defined over a response $\mathbf{y}$ as

$$\mathcal{L}_{\text{TB}}(\mathbf{y}, \mathbf{x}; \theta) = \left( \log \frac{Z(\mathbf{x})\pi_\theta(\mathbf{y} \mid \mathbf{x})}{R(\mathbf{y}; \mathbf{x})} \right)^2. \tag{3}$$

$Z(\mathbf{x})$ is a positive scalar function of the query $\mathbf{x}$, and the response $\mathbf{y}$ is a sequence of tokens. When $\mathcal{L}_{\text{TB}}$ is minimized, $Z(\mathbf{x})$ is the partition function of the posterior (i.e. $Z(\mathbf{x}) = \sum_{\mathbf{y}} R(\mathbf{y}; \mathbf{x})$). Otherwise it serves as a control variate, reducing the variance of TB gradients. Instead of training a value network for $Z(\mathbf{x})$, we use the VarGrad variant of trajectory balance which replaces a learned $Z(\mathbf{x})$ with a K-sample batch estimate [55, 47, 78, 58, 68].

Given $K$ responses $\{\mathbf{y}^{(i,j)}\}_{j=1}^K$ for a query $\mathbf{x}^{(i)}$, a batch estimate of $Z(\mathbf{x}^{(i)})$ can be computed

$$\log \hat{Z}(\mathbf{x}^{(i)}) = \frac{1}{K} \sum_{j=1}^K \left( \log \pi_{\text{ref}}(\mathbf{y}^{(i,j)} \mid \mathbf{x}^{(i)}) - \log \pi_\theta(\mathbf{y}^{(i,j)} \mid \mathbf{x}^{(i)}) + \frac{1}{\beta} r_\phi(\mathbf{y}^{(i,j)}; \mathbf{x}^{(i)}) \right). \tag{4}$$

The (detached) estimate $\hat{Z}$ can be plugged into Eq. 3 for a batch $\mathbf{B} = \{(\mathbf{x}^{(i)}, \mathbf{y}^{(i,j)})_{j=1}^K\}_{i=1}^B$ to get

$$\mathcal{L}_{\text{TB}}^{\text{VarGrad}}(\mathbf{B}; \theta) = \frac{1}{BK} \sum_{i=1,j=1}^{i=B,j=K} \left( \text{STOP-GRAD}[\log \hat{Z}(\mathbf{x}^{(i)})] + \log \pi_\theta(\mathbf{y}^{(i,j)} \mid \mathbf{x}^{(i)}) \right.$$
$$\left. - \log \pi_{\text{ref}}(\mathbf{y}^{(i,j)} \mid \mathbf{x}^{(i)}) - \frac{1}{\beta} r_\phi(\mathbf{y}^{(i,j)}; \mathbf{x}^{(i)}) \right)^2. \tag{5}$$

An important property of the trajectory balance is that it is *off-policy*. During training, $\mathbf{y}$ can be sampled from any distribution with full support [4]. This enables the use of various exploration strategies [54, 30] as well as the use of replay buffers [60, 67]. In the context of fine-tuning language models, this off-policy nature of the objective makes it a natural choice for asynchronous training.

### 3.3 Differences from other RL objectives

We compare the gradients of the VarGrad TB variant above and Proximal RLOO, an off-policy variant of RLOO with off-policy robustness stemming from a (clipped) importance-sampling (IS) ratio [46]. With $K$ samples, the gradient for Proximal RLOO for a sample $\mathbf{y}^{(j)}$ is

$$\nabla \mathcal{J}_{\text{P-RLOO}}(\theta) = \frac{\pi_\theta(\mathbf{y}^{(j)}|\mathbf{x})}{\pi_{ref}(\mathbf{y}^{(j)}|\mathbf{x})} \hat{A}(\mathbf{y}^{(j)}|\mathbf{x}) \nabla_\theta \log \pi_\theta(\mathbf{y}^{(j)}|\mathbf{x}), \tag{6}$$

with advantage $\hat{A}(\mathbf{y}^{(j)}|\mathbf{x}) = r(\mathbf{y}^{(j)}, \mathbf{x}) - \frac{1}{K-1} \sum_{i \neq j} r(\mathbf{y}^{(i)}, \mathbf{x})$, and IS clipping omitted for brevity.

The gradient for $\mathbf{y}^{(j)}$ using the TB objective $\mathcal{J}_{\text{TB}}$ corresponding to the loss in Eq. 5 is, up to a multiplicative constant, equal to the following (derivation given in Appendix A):

$$\nabla \mathcal{J}_{\text{TB}}(\theta) = \hat{A}(\mathbf{y}^{(j)}|\mathbf{x}) \nabla \log \pi_\theta(\mathbf{y}^{(j)} \mid \mathbf{x}), \tag{7}$$

where $\hat{A}(\mathbf{y}^{(j)}|\mathbf{x}) = [r(\mathbf{y}^{(j)}, \mathbf{x}) - \beta \log \frac{\pi_\theta(\mathbf{y}^{(j)}|\mathbf{x})}{\pi_{ref}(\mathbf{y}^{(j)}|\mathbf{x})}] - \frac{1}{K} \sum_i [r(\mathbf{y}^{(i)}, \mathbf{x}) - \beta \log \frac{\pi_\theta(\mathbf{y}^{(i)}|\mathbf{x})}{\pi_{ref}(\mathbf{y}^{(i)}|\mathbf{x})}]$. That is, for on-policy data $\mathbf{y}$, TB$^{\text{VarGrad}}$ is equivalent to mean-baseline REINFORCE [75] and a KL-divergence-regularized reward [82]. However, we use TB on off-policy data, i.e. not sampled from $\pi_\theta$, where this equivalence does not hold.

In practice, we find that performance is sensitive to the coefficient $\beta$, with larger values tending to promote stability, and smaller values tending to promote accuracy improvements. We find it easier to obtain both benefits via coefficient annealing schedules or resetting of the reference policy [36]. Notably, Kimi k1.5 was RL-tuned with an objective that nearly matches TB's and reset the reference policy [65]; Kimi k1.5 deviated from Eq. 7 by excluding the average log probability ratio from its control variate. Brantley et al. [7] also use a related regression objective, learning offline a value function that plays the role of the control variate. Future work may consider adding the (clipped) IS ratio to Eq. 7 to obtain further robustness to off-policy data, or exploring IS clipping alternatives [80].

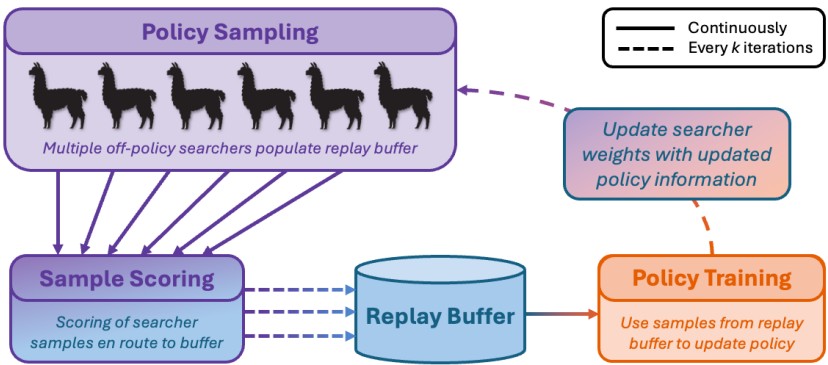

Figure 2: **Fast, scalable LLM post-training with TBA.** Continuously (solid lines), multiple SEARCHER nodes (left) collect trajectories, while a TRAINER node (right) samples from a replay buffer to train the policy off-policy. Periodically (dashed lines), updated policy weights are sent to SEARCHER nodes, and new trajectories are added to the TRAINER node's buffer. This avoids bottlenecks at any given node, which can be 1 or more GPUs, keeping resource utilization high.

## 4 TBA: Fast, Scalable LLM Post-Training

TBA integrates the TB gradient of Eq. 5 into an asynchronous distributed RL framework for post-training language models. Asynchronous RL decouples data generation from model updates, enhancing resource utilization and reducing training duration. At the same time, the off-policy TB objective promotes efficient learning from the off-policy training data induced by asynchrony.

We test two TBA variants. The first implements TBA from scratch by modifying an RLOO trainer class (e.g. of the Hugging Face TRL library [70]) to use Eq. 5, adding a schedule for $\beta$, integrating a replay buffer and sampling approach, and parallelizing training and search to allow async RL – we discuss the details of this variant below. The second variant TBA′ uses the asynchronous PRIME-RL codebase [26], to which we simply add the update rule corresponding to Eq. 7 and reference-policy resetting. TBA′ has fewer tunable knobs than TBA and supports multi-GPU training processes, allowing us to test TB for async RL of LLMs in a simpler setting and with larger models/contexts.

While TBA′ prioritizes ease of use and multi-GPU training, our from-scratch TBA implementation prioritizes speed and search. For example, ensuring the data has a constant level of off-policyness (as PRIME-RL does) can reintroduce the bottleneck that async RL aims to remove, so TBA runs the training and searching processes continuously and independently, only syncing every $k$ steps (the "sync period"). Moreover, every completion, regardless of how stale it is, is accessible in TBA's replay buffer (e.g. via reward-based sampling). As discussed below, $m$ is TBA's probability of sampling the data that is (relatively) the "most on-policy"; i.e., the data added at the most recent sync step.

Figure 2 visualizes our TBA implementation. It leverages overlapping training and search processes, and a buffer to hold trajectories. In particular, TBA uses a single TRAINER node and one or more SEARCHER nodes that collect off-policy data into a shared replay buffer $\mathcal{D}_{\text{global}}$. Here, a node is a computational resource sufficient to perform all the operations needed for training or search – it can be a collection of multiple GPUs or even a subset of 1 GPU. In our TBA experiments, a node is always 1 GPU: e.g., given 16 GPUs, we use 15 SEARCHER nodes and 1 TRAINER node.

Separation of the SEARCHER and TRAINER is highly desirable even in a 2 node (e.g. 2 GPU) cluster because LLM policy rollouts are costly sequential decoding procedures, and training requires only parallel likelihood evaluation of an entire sequence through a single forward pass. Thus, given an objective that can tolerate off-policy asynchronous online training, massive wall clock time speedups can be realized by continuously running training on 1 node without pausing for rollout generation. In the following subsections, we provide more details on our from-scratch TBA implementation.

### 4.1 Scaling data collection with SEARCHER nodes

TBA's SEARCHER nodes each carry a local delayed copy $\pi_{\theta'}$ of the TRAINER policy $\pi_\theta$. To produce policy rollouts, queries $\mathbf{x}$ are sampled from a dataset, and the local policy generates a batch of $K$ responses $\mathbf{y} \sim \pi_{\theta'}(\mathbf{y}|\mathbf{x})$ that are evaluated with the reward model $r_\phi(\mathbf{y}; \mathbf{x})$. Like Noukhovitch

et al. [46], we use vLLM [33] for faster generation. The $(\mathbf{x}, \mathbf{y}, r_\phi(\mathbf{y}; \mathbf{x}))$ tuples are stashed in the SEARCHER's local replay buffer $\mathcal{D}_{\text{local}}$. We also add to the stored tuple the step of the trainer when syncing last occurred, giving us a notion of how off-policy the generated data is – this can later be used by the TRAINER to prioritize sampling from more recent generations.

Every $k$ optimization steps, search and training pause to pull each SEARCHER node's local replay buffer $\mathcal{D}_{\text{local}}$ into the global replay buffer $\mathcal{D}_{\text{global}}$, and to update the searcher's local policy with the trainer's. The global buffer maintains a list of all generated responses and rewards for each query.

A key motivation for asynchrony is its compatibility with rollout scaling for enhanced exploration or increased sequence lengths. For example, we generate $S > K$ samples for a given query – even when only updating the model using $K$ samples per query – to mitigate the lack of diversity caused by the fact that $K$ independent model rollouts are not guaranteed to produce $K$ unique sequences. Relatedly, future work could apply simple off-policy inference techniques in the SEARCHER nodes such as randomly sampling the softmax temperature, or using alternative decoding techniques like beam search. We expect such approaches to particularly aid solution discovery in sparse reward settings.

### 4.2  Asynchronous updates with TRAINER

The TRAINER uses off-policy trajectory balance (Eq. 5) to train the policy on the global replay buffer $\mathcal{D}_{\text{global}}$. We sample a batch of $B$ queries, each with $K$ responses and corresponding rewards:

$$\{\mathbf{x}^{(i)}, \mathbf{y}^{(i,j)}, r_\phi(\mathbf{y}^{(i,j)}; \mathbf{x}^{(i)})\}_{i=1,j=1}^{i=B,j=K} \sim \mathcal{D}_{\text{global}}.$$

We then compute the loss in Eq. 5 and use it to update the policy. We sample with replacement if fewer than $K$ unique samples exist for a given query.

A critical design choice is the strategy for sampling from the replay buffer $\mathcal{D}_{\text{global}}$. The most naive approach is uniform sampling over queries, then uniform sampling over samples associated with the selected query, which may not be optimal if high-reward samples are sparse. Reward prioritization can address this by tilting the sampling distribution toward high-reward sequences; however, focusing solely on high-reward data can lead to mode collapse and reduce policy diversity.

To balance between these concerns, we alternate between two sampling strategies: one prioritizing recency – i.e., whether the trajectory was added to the buffer in the most recent sync step – and another prioritizing rewards. When prioritizing rewards, we consider both a softmax of the reward value (to encourage sampling high reward responses) and a uniform distribution (to encourage sampling high and low reward responses equally). We randomly switch between prioritizing rewards and recency for each query in a batch, with the fraction of queries allocated to each strategy treated as a tunable hyperparameter $m$, which we study in Section 5.4.

## 5  Empirical Evaluation

We evaluate the effectiveness of TBA in three common LLM post-training RL pipelines. Post-training RL for enhancing LLM capabilities—particularly for agentic and reasoning tasks—is a critical but nascent area where baseline approaches require many hours or days. These conventional methods often rely on costly-to-generate on-policy data and thus inefficiently leverage available computing resources. Notably, this inefficiency can become particularly harmful when scaling to larger distributed systems, which may be a necessity for domains with sparse rewards that demand increased sampling. Broadly, we find TBA is a highly-performant, fast, and scalable solution.

### 5.1  Tasks

- **Mathematical reasoning (MR)**: We study the GSM8K task which consists of grade-school level math problems and a binary reward based on exact match for the correct final answer [9]. We adopt the setup from Kazemnejad et al. [29], Noukhovitch et al. [46], using an SFTed RhoMath-1B [35] model as a base for RL post-training.

- **Preference fine-tuning (PFT)**: We consider the task of fine-tuning a language model with a reward function learned from human preference data. Specifically, we study the TL;DR summarization task where the goal is to write short summaries for reddit posts [62]. Following Noukhovitch et al. [46], we consider Pythia [6] as the base model for the policy and the reward models, using the SFTed versions used by Noukhovitch et al. [46].

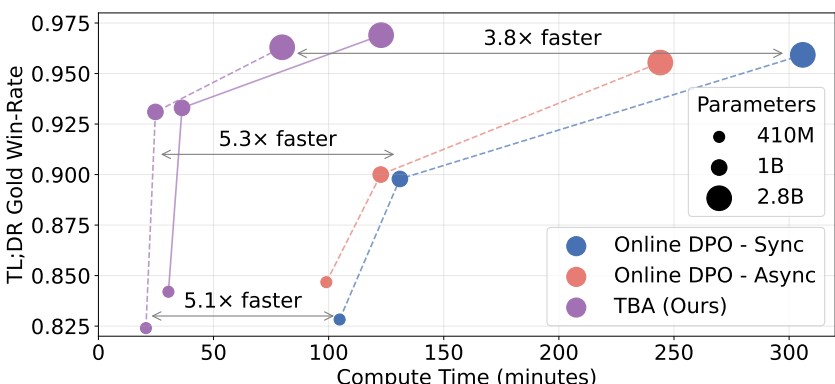

Figure 3: **TBA scales search and improves RL efficiency on the PFT summarization task**. All plotted points use 4xA100 GPUs, but TBA allocates 3 GPUs to search, and Online DPO allocates 1 GPU to search. TBA produces large-scale off-policy data that its trajectory balance objective can leverage, creating massive efficiency benefits. Online DPO baselines taken from Noukhovitch et al. [46]. Dashed and solid lines use 256 and 425 updates, respectively. Appendix B has details.

- **Red-teaming (RT)**: We investigate automated red-teaming, another critical step for LLM post-training. The goal is to discover prompts that elicit harmful responses from a target model, as measured by a toxicity classifier. We follow the setup from Lee et al. [34], applying the same models: our smaller-scale experiments use GPT-2 [52] as an attacker model, GPT-2 (instruction-tuned) as a victim model, and a RoBERTa-based toxicity classifier [69]; our larger-scale experiments use Llama-3.2-1B [41] as an attacker model, Llama-3.1-8B-Instruct as a victim model, and LlamaGuard-3-8B [41] for measuring toxicity, averaging over multiple responses from the victim model.

For all tasks, we study solely the RL post-training components of baselines' workflows. In particular, we start with SFTed checkpoints, then perform RL. Notably, Lee et al. [34] also include a maximum likelihood estimation training phase after RL that uses the buffer produced during RL, which boosts performance but is not investigated here. Using these baselines' codes, we follow their setup and hyperparameters, with deviations noted in Appendix B. For MR and PFT, we implement TBA by augmenting the RLOO trainer of Noukhovitch et al. [46] with our distributed asynchronous RL framework and the TB objective (Equation 5). For RT, we implement TBA by augmenting the trajectory balance trainer of Lee et al. [34] with our distributed asynchronous RL framework.

## 5.2 Metrics and baselines

**MR metrics and baselines.** We follow the evaluation setup of Noukhovitch et al. [46], computing the pass@1 on the GSM8K [9] test dataset with greedy decoding. Prior work [29, 46] applies RL post-training with the GSM8K training set to the SFTed RhoMath-1B [35] model, which initially obtains 40.3% accuracy on the test set. We compare TBA post-training to the methods used in these prior works: **VinePPO** [29], **Online-DPO** [17], (Async) **PPO** [57], and (Proximal) **RLOO** [2]. Additionally, we implement a **GRPO** [59] baseline (see Appendix D.1 for details).

**PFT metrics and baselines.** We follow the evaluation setup of Noukhovitch et al. [46], using the win-rate under a 6.7B "gold" reward model [25] as the primary metric. We additionally report approximate KL distance—approximated by perplexity to match the evaluation of Noukhovitch et al. [46]—between the learned policy and the reference policy. Following Noukhovitch et al. [46], we compare TBA with: **Online-DPO** [17], (Async) **PPO** [57], and (Proximal) **RLOO** [2].

**RT metrics and baselines.** We follow Lee et al. [34], measuring the attack success rate on 1024 sampled prompts for a victim model. We also measure the diversity of these test-time generated prompts by computing the average pairwise cosine distance. We compare against: **SFT**, **PPO** [57] (with the novelty reward from Lee et al. [34]), **REINFORCE** [75, 63], **RLOO** [1], **Online DPO** [53], **GFlowNet (one-actor TB)** [34].

## 5.3 TBA redefines efficiency-performance Pareto frontiers

We hypothesize that, by mixing asynchronous RL with the trajectory balance objective for learning from off-policy data, TBA can (a) reduce resource waste and training time and (b) improve perfor-

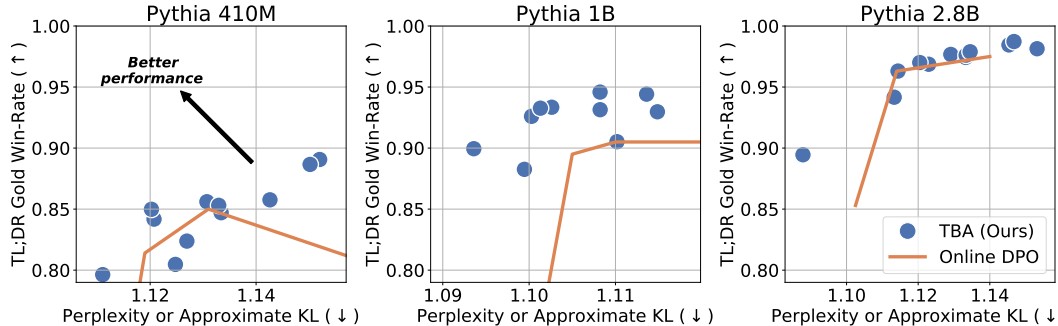

Figure 4: **TBA defines a new KL vs. win-rate Pareto frontier on the PFT summarization task.** The baseline "Online DPO" frontier is created by increasing the degree of off-policyness, starting from on-policy Online DPO, results from [46]. The TBA frontier is created by altering the training steps, searcher count, and KL annealing schedule as described in Appendix B.

Table 1: **TBA outperforms off-policy *and on-policy* baselines in the PFT task (Pythia 410M).** Baselines from Noukhovitch et al. [46]. The final block uses 16 steps off-policy for baselines, and a sync period of 10 for TBA, which corresponds to use of data that is 15 steps off-policy on average.

| Steps/Sync | Metric | Method | | | |
|---|---|---|---|---|---|
| | | Online DPO | (Async) PPO | (Proximal) RLOO | TBA |
| 1 (on-policy) | Perplexity/KL ↓ | 1.13 | 1.14 | 1.13 | — |
| | Win Rate ↑ | 0.85 | **0.86** | 0.82 | — |
| 2 (async) | Perplexity/KL ↓ | 1.13 | 1.14 | 1.13 | — |
| | Win Rate ↑ | 0.85 | 0.84 | 0.83 | — |
| ≈ 15 (async) | Perplexity/KL ↓ | 1.12 | 1.10 | 1.10 | 1.13 |
| | Win Rate ↑ | 0.82 | 0.76 | 0.77 | **0.86** |

mance via scaled generation and ingestion of diverse responses. To test this, we primarily consider compute-matched settings where all methods have access to the same number of GPUs, though we also evaluate TBA with resource scaling. Even in our compute-matched experiments, TBA generates responses relatively rapidly by running asynchronous search on all but one of the available GPUs. Training happens quickly with TBA because it isn't bottlenecked by on-policy generation, and TBA's rapid response generation ensures training on diverse and previously unseen (though off-policy) data.

When tested on established RL problems, an alternative possibility is that TBA will underperform due to its departures from conventional approaches: it is asynchronous, off-policy, reliant on the less-common trajectory balance objective, and makes updates using many responses per query.[1] Indeed, Noukhovitch et al. [46] contemporaneously suggests potential limitations to asynchronous, off-policy RL for LLMs, finding that increasing off-policyness can harm performance metrics like win-rate or exacerbate policy deviations from the reference model. Further, Hou et al. [22] found limited benefits to scaling response generation from 4 to 8 (or 16) responses when using PPO.

We study this question by computing Pareto frontiers for **MR (Figure 1)**, for **PFT (Figures 3 and 4 and Table 1)**, and for **RT (Figure 5 and Table 6)**. See Appendix B for experimental details. Notably, TBA produces results on or beyond the Pareto frontiers of all three tasks at multiple model scales, consistent with our hypothesis that TBA can efficiently train LLMs on off-policy data.

Speed is vastly improved with TBA training, which proceeds entirely asynchronously without training-bound or generation-bound processes – the only non-training time occurs briefly every $k$ steps. In the compute-matched MR experiments (Figure 1), TBA speeds up the training of the only method with comparable performance (VinePPO) by nearly $50\times$, while improving accuracy by $1.8\%$ and speed

---

[1]We compute the TBA loss with more samples per query than analogous approaches like RLOO (e.g., 20 samples versus 4) to reduce the variance of the gradient of the TB objective estimate (Equation 5) that we optimize. We use fewer queries per batch to keep the batch size small despite this response scaling.

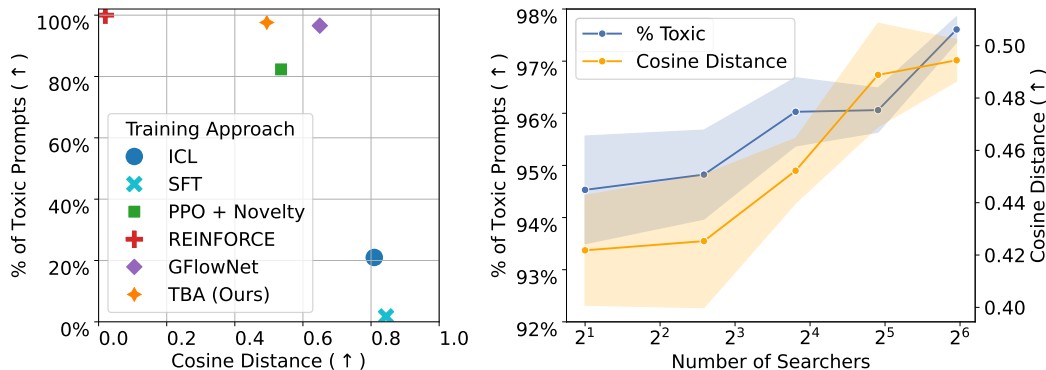

Figure 5: **TBA reaches the RT diversity-toxicity Pareto frontier and improves as search is scaled**. **(Left)** On the GPT-2 automated red-teaming task of Lee et al. [34], TBA produces results on the diversity vs. toxicity Pareto frontier in less training time. Baselines taken from Lee et al. [34]. **(Right)** Each searcher uses one V100 GPU for generating attacks. We report means and standard errors from multiple runs of the automated red-teaming task with GPT-2 at each searcher/GPU count.

by $1.5\times$ relative to the speed-optimized asynchronous DPO baseline [46]. In the compute-matched PFT experiments (Figure 3), TBA produces $\approx 5\times$ speedups over speed-optimized asynchronous DPO baselines [46], and it even outperforms baselines that train with on-policy data (Table 1). In the non-compute-matched automated red-teaming experiments (Table 6), TBA gives $\approx 7\times$ speedups for GPT-2 and Llama 3.2 1B compared to the non-distributed, synchronous GFlowNet baseline [34]. These results suggest that TBA is an effective, parallel, and scalable search framework for distributed learning, offering substantial speed-ups while remaining competitive with leading approaches.

## 5.4 Does off-policyness hurt performance?

The results in the prior section are perhaps surprising given evidence of off-policyness's harmfulness to RL post-training [46]. Thus, we investigate the effect of off-policyness with TBA by studying its related hyperparameters, which we first review here. The fraction $m$ controls how often sampling attempts to approximate an "on-policy" distribution. When $m = 1$, training occurs exclusively with samples added in the most recent sync step, while $m = 0$ corresponds to selecting data without regard for how recently it was generated (e.g., using reward weighting). Additionally, recall that parameters and buffer data are shared between searchers and trainers every $k$ training steps: since TBA runs exploration and training in parallel, TBA trains off-policy even when $m = 1$ and $k = 1$. Specifically, with probability $m$, TBA selects a sample from the central replay buffer that is at most $2k - 1$ updates off-policy. With probability $1 - m$, TBA samples data produced by the model at any point, which can be as off-policy as the number of training steps.

To understand the effect of increasing off-policyness on TBA, we test the effect of modifying $m$ for the **MR (see Figure 7)** and **PFT** tasks. For PFT, we train Pythia-410M on the TL;DR dataset with three values of $m$ $(0.4, 0.5, 0.6)$ keeping all other parameters constant. We found that win rate fluctuated, with $m = 0.4$ corresponding to the lowest win rate $(0.67)$, and $m = 0.5$ and $m = 0.6$ attaining higher win rates of $0.82$ and $0.8$, respectively. These results show that higher values of $m$ generally lead to a higher win rate, reinforcing the idea that on-policy updates are in fact the most effective. However, for reasonably high values of $m$, incorporating more off-policy data does not significantly degrade performance and, in some cases, may even provides benefits. Regardless, our findings for both MR and PFT further support the idea that we can perform massively distributed training that works well with off-policy updates, as long as recent samples are thrown into the mix.

Importantly, the choice of reinforcement learning (RL) algorithm is crucial in determining how effectively we can leverage off-policy updates. TBA is a fully off-policy compatible algorithm, and as shown in Figure 4, it significantly outperforms asynchronous Online DPO, even in the latter's most on-policy setting. Noukhovitch et al. [46] identified Online DPO as the best-performing off-policy algorithm in their experiments (see Table 1), making it notable that TBA improves upon this.

## 5.5 Discovery of high-reward samples via scaling search

Beyond improvements at a given level of compute, we also observe improvements when we scale the amount of total compute, showing TBA's promise for large-scale distributed RL. In our asynchronous setup, we find that adding more searchers consistently improves the attack success rate and diversity for **RT (see Figure 5, right)**.

This improvement likely stems from having more searchers exploring different regions of the solution space simultaneously, enabling more effective discovery of high-reward samples. Moreover, asynchronous updates introduce opportunities for beneficial randomness, and thus potential expansion of the search coverage in the combinatorial space of language. Interestingly, we also find some evidence for scaling's helpfulness in **PFT (see Figure 9)**.

Relatedly, in **Table 6**, we show we can trade attack toxicity for greater attack diversity by scaling the maximum buffer size to retain more off-policy data. This RT experiment uses Llama 3.2 1B models.

## 6 Scaling TBA for Larger Models

As described in Section 4, TBA′ allows us to test a simplified version of TBA and to train with larger models/contexts. Relative to TBA, TBA′ has fewer knobs to tune: it does not have a decay schedule for $\beta$, and PRIME-RL does not sample from a buffer, using a constant level of off-policyness instead. A TBA′ configuration entails a reference policy reset interval $\rho$, and a constant $\beta$. We use $\rho = 50$, $\beta = 0.005$. In **Figure 6**, we find further evidence that the TB objective can enhance async RL training of LLMs, particularly when data is highly off-policy. See Table 4 for experimental hyperparameters.

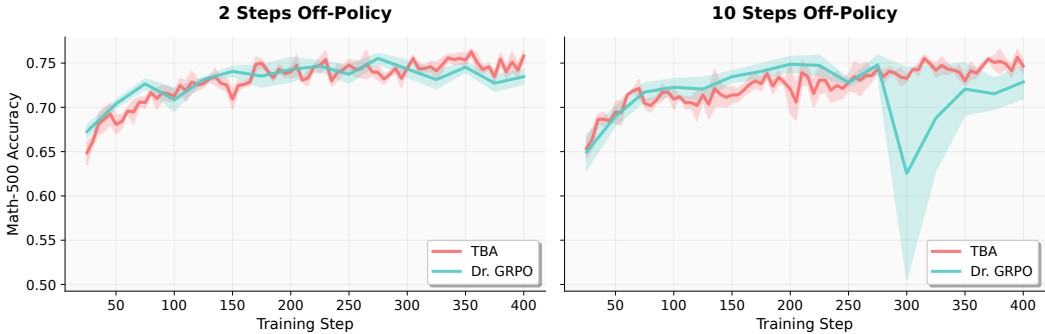

Figure 6: **TBA′ vs. Dr. GRPO.** When training Qwen 2.5 7B base [66] on the MATH dataset [18], we find that TBA performs well relative to state-of-the-art methods, especially in highly off-policy settings (right).

## 7 Discussion

In this work, we introduced TBA, a novel post-training method for large language models (LLMs) that combined an off-policy reinforcement learning (RL) objective with distributed asynchronous search. By decoupling searcher and trainer nodes, our approach enabled efficient distributed training and avoided bottlenecks, leading to significant performance gains in post-training tasks such as mathematical reasoning, automated red-teaming, and RLHF. We expect that our highly parallelizable and performant framework for RL post-training can be extended to other valuable tasks, including self-improvement training [77, 23, 15], search-based reasoning [71], as well as the emerging paradigm of training LLMs to reason with RL [16].

**Limitations** The trajectory balance objective can suffer from high gradient variance as it operates on the trajectory level. We addressed this by sampling more responses per query. Future work can leverage learning partial energy functions [38, 79] to balance bias and variance during policy updates.

**Broader Impact** Improving RL training strategies for LLMs can enhance their usefulness across domains but also carries risks associated with misuse, reward misspecification, and unintended generalization. Careful evaluation and responsible deployment are essential as these methods scale.

## Acknowledgments and Disclosure of Funding

We thank Michael Noukhovitch and Haizhong Zheng for helpful feedback.

The authors acknowledge funding from CIFAR, NSERC, IVADO, and Samsung. MJ is supported by a FRQNT Doctoral Fellowship (https://doi.org/10.69777/366694).

Prepared by LLNL under Contract DE-AC52-07NA27344 and supported by the LLNL-LDRD Program under Project No. 24-ERD-058 (LLNL-CONF-2003261). This manuscript has been authored by Lawrence Livermore National Security, LLC under Contract No. DE-AC52-07NA27344 with the U.S. Department of Energy. The United States Government retains, and the publisher, by accepting the article for publication, acknowledges that the United States Government retains a non-exclusive, paid-up, irrevocable, world-wide license to publish or reproduce the published form of this manuscript, or allow others to do so, for United States Government purposes.

This research used resources of the National Energy Research Scientific Computing Center (NERSC), a Department of Energy Office of Science User Facility using NERSC awards ASCR-ERCAP0032802 and ASCR-ERCAP0032812.

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

## A TBA Gradient Analysis

Below, we reproduce for convenience the TBA loss $\mathcal{L}_{\text{TB}}^{\text{VarGrad}}$ from Eq. 5, then provide the gradient of the corresponding maximization objective $\mathcal{J}_{\text{TB}}$.

$$\mathcal{L}_{\text{TB}}^{\text{VarGrad}}(\mathbf{B}; \theta) = \frac{1}{BK} \sum_{i=1,j=1}^{i=B,j=K} \left( \text{STOP-GRAD}[\log \hat{Z}(\mathbf{x}^{(i)})] + \log \pi_\theta(\mathbf{y}^{(i,j)} \mid \mathbf{x}^{(i)}) \right.$$
$$\left. - \log \pi_{\text{ref}}(\mathbf{y}^{(i,j)} \mid \mathbf{x}^{(i)}) - \frac{1}{\beta} r_\phi(\mathbf{y}^{(i,j)}; \mathbf{x}^{(i)}) \right)^2. \tag{8}$$

We first multiply everything inside the square operation of Eq. 5 by $-1$, which is equivalent to multiplying the entire equation by 1 and gives

$$\mathcal{L}_{\text{TB}}^{\text{VarGrad}}(\mathbf{B}; \theta) = \frac{1}{BK} \sum_{i=1,j=1}^{i=B,j=K} \left( \log \pi_{\text{ref}}(\mathbf{y}^{(i,j)} \mid \mathbf{x}^{(i)}) - \log \pi_\theta(\mathbf{y}^{(i,j)} \mid \mathbf{x}^{(i)}) + \right.$$
$$\left. \frac{1}{\beta} r_\phi(\mathbf{y}^{(i,j)}; \mathbf{x}^{(i)}) - \log \hat{Z}(\mathbf{x}^{(i)}) \right)^2. \tag{9}$$

The gradient with respect to $\theta$ then becomes

$$\nabla \mathcal{L}_{\text{TB}}^{\text{VarGrad}}(\mathbf{B}; \theta) = \frac{1}{BK} \sum_{i=1,j=1}^{i=B,j=K} -2 \left( \log \pi_{\text{ref}}(\mathbf{y}^{(i,j)} \mid \mathbf{x}^{(i)}) - \log \pi_\theta(\mathbf{y}^{(i,j)} \mid \mathbf{x}^{(i)}) + \right.$$
$$\left. \frac{1}{\beta} r_\phi(\mathbf{y}^{(i,j)}; \mathbf{x}^{(i)}) - \log \hat{Z}(\mathbf{x}^{(i)}) \right) \nabla \log \pi_\theta(\mathbf{y}^{(i,j)} \mid \mathbf{x}^{(i)}), \tag{10}$$

and we can turn this into a maximization problem instead of a minimization problem by multiplying the right side by $-1$:

$$\nabla \mathcal{J}_{\text{TB}}^{\text{VarGrad}}(\mathbf{B}; \theta) = \frac{1}{BK} \sum_{i=1,j=1}^{i=B,j=K} 2 \left( \log \pi_{\text{ref}}(\mathbf{y}^{(i,j)} \mid \mathbf{x}^{(i)}) - \log \pi_\theta(\mathbf{y}^{(i,j)} \mid \mathbf{x}^{(i)}) + \right.$$
$$\left. \frac{1}{\beta} r_\phi(\mathbf{y}^{(i,j)}; \mathbf{x}^{(i)}) - \log \hat{Z}(\mathbf{x}^{(i)}) \right) \nabla \log \pi_\theta(\mathbf{y}^{(i,j)} \mid \mathbf{x}^{(i)}). \tag{11}$$

We multiply the right hand side by $\frac{\beta}{2}$, rescaling the gradient without changing the minimizer. Also, we define $\bar{\text{KL}}^{(i)} = \frac{1}{K} \sum_{j=1}^{K} \hat{\text{KL}}^{(i,j)} = \frac{1}{K} \sum_{j=1}^{K} \log \pi_\theta(\mathbf{y}^{(i,j)} \mid \mathbf{x}^{(i)}) - \log \pi_{\text{ref}}(\mathbf{y}^{(i,j)} \mid \mathbf{x}^{(i)})$ and define $\bar{r}^{(i)} = \frac{1}{K} \sum_{j=1}^{K} r^{(i,j)} = \frac{1}{K} \sum_{j=1}^{K} r_\phi(\mathbf{y}^{(i,j)}; \mathbf{x}^{(i)})$ to rewrite Eq. 4 as $\log \hat{Z}(\mathbf{x}^{(i)}) = \frac{1}{\beta} \bar{r}^{(i)} - \bar{\text{KL}}^{(i)}$:

$$\nabla \mathcal{J}_{\text{TB}}(\mathbf{B}; \theta) = \frac{1}{BK} \sum_{i=1,j=1}^{i=B,j=K} A^{(i,j)} \nabla \log \pi_\theta(\mathbf{y}^{(i,j)} \mid \mathbf{x}^{(i)}), \tag{12}$$

where $A^{(i,j)} = (r^{(i,j)} - \bar{r}^{(i)}) - \beta(\hat{\text{KL}}^{(i,j)} - \bar{\text{KL}}^{(i)})$. In sum, TBA appears to update model weights using a REINFORCE algorithm with a mean-reward baseline [75] and a KL-divergence-regularized reward [82]. However, note that $\hat{\text{KL}}^{(i,j)}$ is only an estimate of the KL divergence in the on-policy case because off-policy data is not sampled from $\pi_\theta$ by definition, causing loss of equivalence to mean-baseline REINFORCE with a KL-regularized reward in the asynchronous settings that TBA operates in. Deviating further, TBA may also reset the reference policy similar to Liu et al. [36].

# B  TBA Experiment Details

As discussed in Sections 4 and 5.4, TBA introduces new hyperparameters. Most notably, these include (1) the sync period $k$, which is the number of training steps between two successive model-buffer synchronizations; and (2) the most-on-policy probability $m$, which is the probability of sampling the data that is most on-policy – i.e., the samples that were added to the buffer during the most recent synchronization. When listing hyperparameter values, we clarify those that are specific to TBA and include references to their discussion/visualization in the text.

For MR and PFT, we implement TBA by building on the RLOO trainer class of the Hugging Face TRL library [70]. For MR and PFT, we make modifications to the baseline hyperparameters as shown in Table 2 and Table 3, respectively.

Our RT implementation of TBA built on the TB trainer used in Lee et al. [34], and we largely utilize their code's default hyperparameters, with differences noted in Appendix B.3.

## B.1  GSM8K Mathematical Reasoning (MR)

All baselines and TBA results use the same starting point, a RhoMath-1B [35] model SFTed on GSM8K training data by Kazemnejad et al. [29] – "realtreetune/rho-1b-sft-GSM8K" on Hugging Face. The baseline model achieves 40.3% test accuracy. For the VinePPO baseline, training time is estimated using their reported 380 seconds per training step and 650 steps for GSM8K training [29]. We implement the GRPO baseline using its Hugging Face TRL trainer, see Appendix D.1 for details.

Table 2: TBA Training Hyperparameters for the GSM8K MR task.

| Hyperparameter | Value | Reference |
|---|---|---|
| Model | Rho-1B SFT on GSM8K | |
| Learning Rate | $1 \times 10^{-5}$ | |
| Learning Rate Schedule | Warmup Stable Decay | |
| Learning Rate Warmup Steps | 50 | |
| Learning Rate Stable Steps | 450 | |
| Learning Rate Decay Steps | 500 | |
| Generation Temperature | 0.7 | |
| Max Prompt Token Length | 512 | |
| Response Length | 512 | |
| Number of Prompts per Batch | 7 | |
| Number of Completions per Prompt | 20 | $K$ in Section 4.1 |
| Batch Size (effective) | 140 | |
| Number of Training Steps | 1000 | |
| Total Prompts Seen | 7000 | |
| Total Episodes | 140000 | |
| *TBA-specific hyperparameters* | | |
| Beta (KL coefficient) Initial Value | 0.012 | $\beta$ in Equation 5 |
| Beta Final Value | 0.004 | $\beta$ in Equation 5 |
| Beta Linear Decay Schedule End Step | 500 | $\beta$ in Equation 5 |
| Number of Samples per Prompt | 24 | $S$ in Section 4.1 |
| Most-On-Policy Sampling Probability | 0.95 | $m$ in Section 5.4 |
| Sync Period | 2 | $k$ in Section 5.4, Figure 2 |
| Number of Searchers | 3 | Searchers in Figure 2 |
| Number of Trainers | 1 | Trainer in Figure 2 |
| Reward-Based Sampling Prioritization | Uniform | Section 4.2 |
| Initial Completions in Buffer | 500 | Buffer at Step 0 in Figure 2 |

**In Figure 1, TBA uses the settings in Table 2 with a few modifications** to shorten training time by an additional 30%, down to 82 minutes on 4xA100 GPUs. In particular, we observed in initial testing (see Appendix D) that using the hyperparameters in Table 2 led to no improvement in performance for the final 300 steps. Thus, we shrank the training duration from 1000 steps to 700 (98000 episodes with batch size 140). Additionally, we made the following modifications to attempt to reduce the variance of the shortened run: 350 stable learning rate steps (down from 450), 0.014 Beta Initial Value

Table 3: **TBA Training Hyperparameters for the TL;DR PFT task.** For the PFT task, we accelerate the decay of Beta. In particular, the Beta Linear Decay Schedule End Step is set to be half the number of training steps, but we abruptly end this decay and set Beta to its final value at one eighth the number of training steps (e.g., step 32 for 256 steps). This has the effect of trading off KL/perplexity, which we found to be relatively low with our TBA setup, for win rate.

| Hyperparameter | Value | Reference |
|---|---|---|
| Model | Pythia SFTed on TL;DR | |
| Learning Rate | $3 \times 10^{-6}$ | |
| Learning Rate Schedule | Linear | |
| Generation Temperature | 0.7 | |
| Max Token Length | 1024 | |
| Max Prompt Token Length | 512 | |
| Response Length | 128 | |
| Number of Prompts per Batch | 8 | |
| Number of Completions per Prompt | 20 | $K$ in Section 4.1 |
| Batch Size (effective) | 160 | |
| Number of Training Steps | 256 | |
| Total Prompts Seen | 2048 | |
| Total Episodes | 40960 | |
| *TBA-specific hyperparameters* | | |
| Beta (KL coefficient) Initial Value | 1 | $\beta$ in Equation 5 |
| Beta Final Value | 0.05 | $\beta$ in Equation 5 |
| Beta Linear Decay Schedule End Step | See caption | $\beta$ in Equation 5 |
| Number of Samples per Prompt | 20 | $S$ in Section 4.1 |
| Most-On-Policy Sampling Probability | 0.5 | $m$ in Section 5.4 |
| Sync Period | 10 | $k$ in Section 5.4, Figure 2 |
| Number of Searchers | 3 | Searchers in Figure 2 |
| Number of Trainers | 1 | Trainer in Figure 2 |
| Reward-Based Sampling Prioritization | Softmax of Score | Section 4.2 |
| Initial Completions in Buffer | 10000 | Buffer at Step 0 in Figure 2 |

(up from 0.012). We ran this experiment three times – obtaining performances of 55.8%, 53.9%, and 54.1% – and reported the mean accuracy 54.6% in Figure 1.

**Limitations and future work**    The 700-step result we show in Figure 1 has standard error 0.6%, which is a little more variance than what we observe in the original 1000 step setup (the blue line in the bottom left plot of Figure 7 shows the mean and standard errors for the 1000 step runs). Future work could further explore variance reduction approaches/hyperparameters for shorter runs (and for TBA/RL more generally).

One way to deal with variance is to choose a checkpoint based on a held out validation set, a strategy used to produce the VinePPO result [29]. We do not use this approach but note that our results would likely benefit significantly ($\approx 1\%$) from it. In particular, each of our runs tends to produce (at some point during training) a higher performance than the performance at the final step – this is expected if you consider that the model performance varies around the average value it converges to towards the end of training (e.g., see again the blue line in the bottom left plot of Figure 7). Despite its being lower than the maximum performance achieved at any point during training, we report this final step performance, which is what a user of TBA could expect to obtain without applying techniques like early-stopping.

## B.2    TL;DR Preference Fine Tuning (PFT)

For **PFT in Figure 3**, we use the settings in Table 3 as well as a longer-duration run with 425 updates.

For **PFT in Figure 4**, we create the TBA frontier by modifying the training steps, searcher count, and beta linear decay schedule shown in Table 3. We train for 256, 425, 625, 725, and 825 steps, and we search with 2, 3, 4, and 7 searcher nodes. We did not notice a significant pattern in performance when changing searcher count, but we found a tendency for higher KL/perplexity values and win-rates with

more training steps (an expected tradeoff from optimizing the policy further). We noticed that the 2.8B model did not create a wide range of KL/perplexity values with these initial runs, so we also performed runs at that model size (with 825 steps, and with 2 and 4 searchers) using a less rapid beta linear decay schedule (reaching the beta final value at step 140 instead of step 104). This schedule change had the effect of reducing the KL/perplexity and win-rate (another expected tradeoff).

For **PFT in Table 1**, we use the settings in Table 3, except we train for 625 steps (100000 episodes) because we found use of more steps tended to improve win rate without a significant increase in perplexity (approximate KL). Additionally, we only use 2 searchers in this run. See Figure 9 for a depiction of the effects of step count and searcher count.

**Limitations and future work**   All of our PFT results were run in 32-bit precision and without DeepSpeed, which was used by baselines we compared against [46]. Lower precision and such training acceleration packages could further improve the speedups we show. Relatedly, for our 2.8B runs, we used gradient checkpointing to fit more examples into a micro batch, which led to slowdowns at this scale (i.e., we only have a 3.8x speedup over the baseline in this setting). We leave the optimization of our framework with appropriate packages to future work. Finally, we used a large number (10000) of initial completions in the buffer, and future work should confirm that a smaller number (e.g. 1000) works equally well – note that a small number worked well for GSM8K.

### B.3   Automated Red Teaming (RT)

Unlike our MR and PFT implementations, our RT implementation uses the code of Lee et al. [34] as a baseline and thus does not follow the Hugging Face TRL trainer style. We discuss hyperparameters in the context of their trajectory balance trainer approach below. We adopt their code largely as it is, with the exception that our TBA implementation uses larger replay buffers than Lee et al. [34] to accommodate our scaled search for trajectories. Additionally, unlike Lee et al. [34], we remove the oldest samples when the buffers fill up as opposed to the lowest reward samples.

For the **Llama results in Table 6**, our hyperparameter choices largely follow those of Lee et al. [34]. We train for 5000 steps with batch size 128. For temperature sampling, we use a low and high of 0.7 and 2.0, respectively. We use a reward schedule horizon of 1000. The language model schedule end is 1.2, and its schedule's horizon is 2000. We prioritize sampling based on reward. We use Beta 0.05. We use sync period ($k$) 10. We use 6 searchers. We use most-on-policy probability ($m$) 0.5 and 0.6 in combination with maximum buffer sizes 150000 and 130000, respectively. By having a smaller maximum buffer size and larger $m$, the latter setting is expected to focus on more recently generated data and prioritize reward over diversity, which is what we observe in Table 6.

For the **GPT2 results in Figure 5 and Table 6**, our hyperparameter choices again largely follow those of Lee et al. [34]. We train for 5000 steps with batch size 128. We use Beta 0.05. We use sync period ($k$) 10. We use most-on-policy probability ($m$) 0.5. We cap the maximum buffer size at 100000 in all experiments; we additionally prevent the buffer from growing past its size as of step 4000 to encourage focusing on more recent data (larger most on policy probabilities $m$ may provide a similar effect). We test searcher counts 2, 6, 14, 30, 62, and we use 3 runs per configuration for error bars and averages.

**Limitations and future work**   In Table 6, there is a slowdown as the number of searchers scales. We believe this is largely addressed by a newer version of our code that uses more efficient buffer communication, but we have not re-run these results yet to confirm this. In any case, developing more efficient TBA implementations is an interesting direction for future work given TBA's ability to effectively leverage large-scale data generation.

## C   TBA′ Experiment Details

In Figure 6, we compare TBA′ to the Dr. GRPO [37] implementation in PRIME-RL [26]. The hyperparameters used are given in Table 4.

Table 4: TBA$'$ Training Hyperparameters for the MATH-500 MR task.

| Hyperparameter | Value | Reference |
|---|---|---|
| Model | Qwen 2.5 7B Base | |
| Learning Rate | $1 \times 10^{-6}$ | |
| Generation Temperature | 1.0 | |
| Max Prompt Sequence Length | 1024 | |
| Max Model Length | 3072 | |
| Number of Prompts per Batch | 32 | |
| Number of Completions per Prompt | 16 | $K$ in Section 4.1 |
| Batch Size (effective) | 512 | |
| Number of Training Steps | 400 | |
| Total Prompts Seen | 12800 | |
| Total Episodes | 204800 | |
| *TBA$'$-specific hyperparameters* | | |
| Beta (KL coefficient) | 0.005 | $\beta$ in Equation 5 |
| Reference policy reset period | 50 | $\rho$ in Section 6 |

# D  GSM8K Ablation Studies

We adopted the hyperparameters for our GSM8K result in Figure 1 based on a series of trial experiments centered around the hyperparameters shown in Table 2. In Figure 7, we show the effects of changing what we found to be key hyperparameters. We note the following observations about TBA hyperparameter effects on GSM8K.

1. Unlike PFT, it was important for $m$ to be somewhat large for the GSM8K MR task (Figure 7, top left). Similarly, syncing more frequently was beneficial (Figure 7, top right). Together, these results suggest that GSM8K performance is more sensitive to off-policyness than performance on other tasks (e.g., PFT).

2. We found that the WSD schedule could add stability (Figure 7, bottom right).

3. Using smaller Beta Final Values tended to improve performance (Figure 7, bottom left), but training became unstable around 0.003. This suggests a tradeoff between stability and accuracy for GSM8K that is mediated by the KL coefficient $\beta$.

4. A batch size of 140 did not provide significantly better or worse results than larger batch sizes in initial testing, but smaller batch sizes allow for faster training steps, motivating our use of 140.

Subsequently, we considered increasing the number of completions per prompt $K$ from 20 to 40. In Table 5, we see that this reduces variance and improves performance. Critically, this result does not require longer training times, as we simply reduce the number of unique prompts per batch to keep the batch size constant.

Table 5: **On GSM8K, doubling $K$ improves TBA's accuracy and variance.**

| K | Test Accuracy | Standard Deviation |
|---|---|---|
| 20 | 54.61 | 0.85 |
| 40 | **54.89** | **0.49** |

## D.1  GRPO Baseline Creation and Training Dynamics Comparison

We implement the GRPO baseline using Hugging Face TRL (version 0.17.0). Relative to the baseline GRPOConfig hyperparameters, we only change the following: Num Train Epochs = 8, Gradient Accumulation Steps = 5, and Per Device Train Batch Size = 32. For training speed acceleration, we use DeepSpeed with ZeRO Stage = 2. We find these settings lead to good GPU memory usage and utilization on the same 4xA100 (80 GB) node that we used for our TBA experiments. Moreover, the batch size (640), total training epochs (8), and number of optimization steps (744) are based on

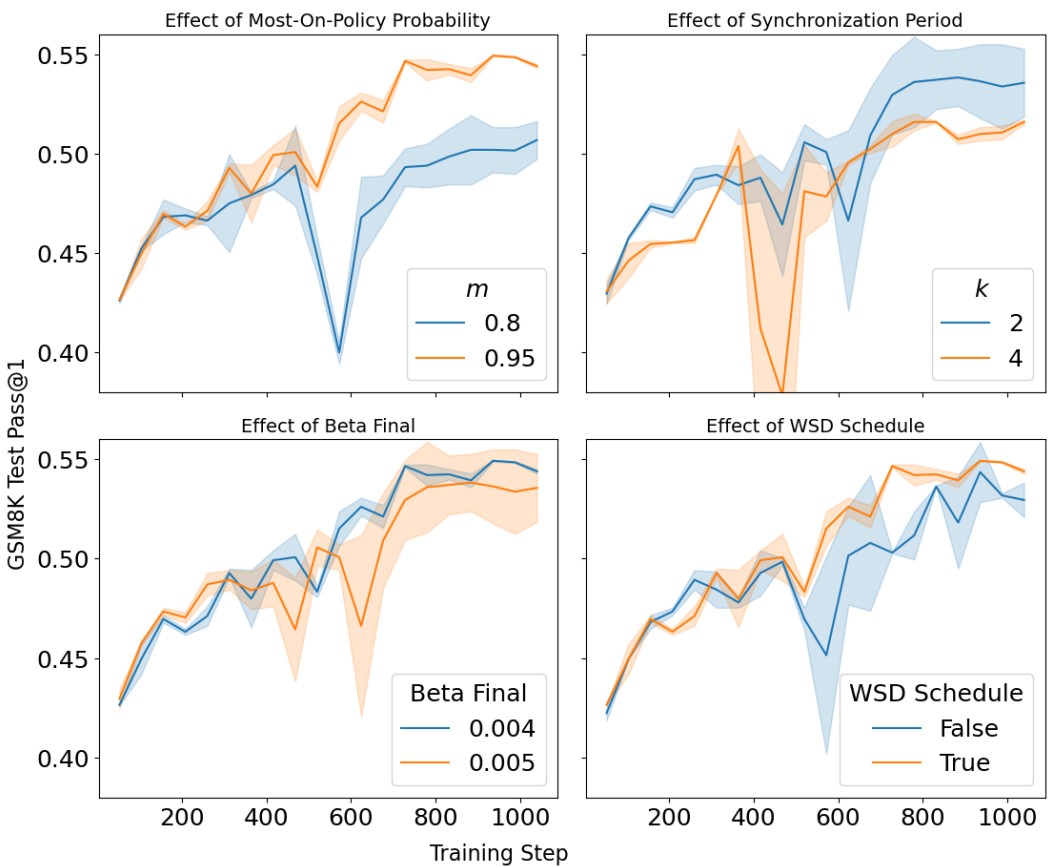

Figure 7: **GSM8K ablation studies.** All experiments begin with the base hyperparameters listed in Table 2 and make the depicted modifications, except when studying the synchronization period $k$ in the top right plot (where we use Beta Final Value $0.005$ because $0.004$ led to instability with $k = 4$). We report the mean and standard error from 2 runs of each configuration.

settings chosen for RL with VinePPO [29] – the paper that created the SFT baseline that we apply GRPO to in our GSM8K experiment (see Figure 1).

To arrive at the GRPO result in Figure 1, we explored several implementation variations of synchronous GRPO. Interestingly, we also tried async GRPO by replacing the loss function used in our async TBA code with GRPO's. We tested several hyperparameter settings with async GRPO inside TBA's async setup – raising the KL coefficient, lowering the learning rate, and decreasing off-policyness – but did not find a setting that provided stable learning.

The average training data score (GSM8K correctness) by training step for our final (synchronous) GRPO implementation and a random TBA replicate are shown in Figure 8. The final model from this GRPO run is used to compute the test performance shown in Figure 1, 49.05%. This TBA run achieves 54.51% test accuracy, a 5% improvement despite the similar training performance levels.

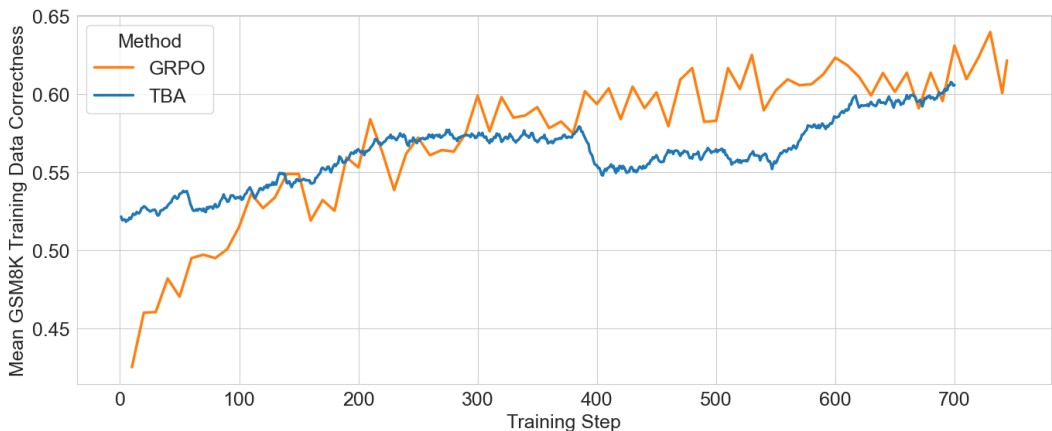

Figure 8: **Training dynamics of GRPO and TBA runs.** The TBA performances are heavily smoothed through an exponential moving average (without smoothing, the TBA curve oscillates such that the trend is harder to see), whereas the GRPO performances are not smoothed.

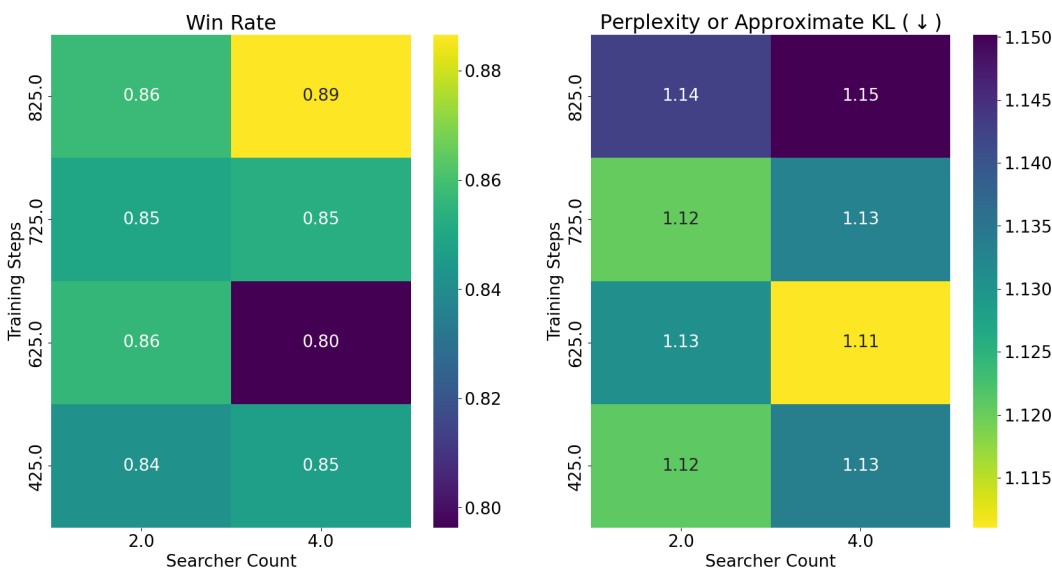

Figure 9: **TL;DR ablation studies.** All experiments begin with the base hyperparameters listed in Table 3 then make the depicted modifications. More searcher nodes and more training steps tend to improve win rate at the cost of higher perplexity.

# E  TL;DR Ablation Studies

With TBA, we take many more optimization steps per hour of training than competing methods take. Accordingly, we sought to understand how taking more steps or using more searchers (to reach compute-parity with competing methods) affects performance. As shown in Figure 9, win rate tends to improve with increased compute through changes to these variables, though perplexity suffers as expected when we over-optimize the policy (in the case of step scaling). It is not clear why scaling the searcher count seems to have an effect similar to step scaling. However, there is a notable mechanism through which searcher scaling could have an effect: using more searchers should reduce the probability of selecting a training prompt that's been seen before when sampling off-policy (because scaling the searchers scales the number of unique prompts with completions added to the

buffer).[2] However, the effect size is small and inconsistent enough to suggest this searcher-scaling trend (in the context of PFT) needs further investigation before it's confirmed.

# F   Red-Teaming Additional Results

Table 6: **TBA speeds up the wall-clock time required to reach the Pareto frontier for the red-teaming task**. The GFlowNet performances are taken from Lee et al. [34], while the training speeds are computed by us with their code. With the GPT-2 models, TBA performance improves with searcher count. With the Llama models, we trade attack toxicity for attack diversity by scaling the TBA buffer's maximum size from 130,000 (penultimate row) to 150,000 samples (final row), retaining more off-policy data.

| Attacker / Victim Model | Training Method | Hardware | Time (h) | Speedup | Cosine Distance | % Toxic Prompts |
|---|---|---|---|---|---|---|
| GPT-2 / GPT-2 + SFT | GFlowNet - Sync | 1×V100 | 11.9 | 1x | 0.65 | 96.6 |
| | TBA (Ours) | 4×V100 | 1.7 | 7x | 0.42 | 94.5 |
| | TBA (Ours) | 16×V100 | 2.5 | 4.8x | 0.45 | 96 |
| | TBA (Ours) | 64×V100 | 2.9 | 4.1x | 0.49 | 97.6 |
| Llama 3.2 1B / Llama 3.1 8B - Instruct | GFlowNet - Sync | 1×A100 | 37.4 | 1x | 0.32 | 100.0 |
| | TBA (Ours) | 8×A100 | 5.7 | 6.6x | 0.35 | 98.1 |
| | TBA (Ours) | 8×A100 | 5.7 | 6.6x | 0.37 | 94.8 |

---

[2]Notably, the effect of scaling search is different in the case of RT, where there is a single prompt and scaling searchers makes it more likely that higher-reward samples are found and trained on. Here, with PFT, scaling search means that more prompts per second have a set of completions generated for them, but any given prompt is not expected to have higher-reward samples as a result of scaling the searcher count – this is an implementation choice and could be changed to gain the effect that scaling searchers has with RT, however.

