# OpenReview forum: "Trajectory Balance with Asynchrony: Decoupling Exploration and Learning for Fast, Scalable LLM Post-Training"
_NeurIPS.cc/2025/Conference — NeurIPS 2025 poster_

### Official Review · Reviewer_ZkfV · 2025-06-22

**Clarity:** 3
**Significance:** 3
**Originality:** 3
**Rating:** 6
**Confidence:** 4

**Summary:**

I summarize the paper as follows. This paper proposes Trajectory Balance with Asynchrony (TBA), a reinforcement learning (RL) framework for post-training large language models. In detail, TBA decouples data generation and policy updates using asynchronous, off-policy training via the trajectory balance objective, kind of diversity-seeking RL objective. It achieves 4x to 50x speedups and improves performance on tasks like mathematical reasoning, preference tuning, and red-teaming compared to existing methods.

**Questions:**

See the weakness part:

**(A) There requires an efficiency discussion in sequence selection.** As mentioned in **Line169-172**, the proposed method generates more samples for each query and selects the subset of generated sequences based on recency and rewards to update the model, this inevitably brings additional memory and inference cost. When the base model is large enough, e.g., >7B params, the efficiency can be degraded.

Based on the above analysis, reinforcement finetuning LLMs is expensive in computations, memories and reward annotations. As far as I know, there exist some amortization methods [2-4] to approximately score the sequence reward or predict the preference of tasks in RL. Hence, *I suggest to add a few discussion sentences regarding the chance of marrying amortized evaluation [3,4] with TBA to improve efficiency and robustness in the future* in **Limitation part Line 325-Line 327**.

**(B) I want to see the discussion between GRPO [5] in DeepSeek-R1 and asynchronous off-policy methods such as TBA.** The analysis is not from quantitative results but some technical discussions. Will removing out the critic be better? I guess this is up to memory constraints.

**Ethical Concerns:**

["NO or VERY MINOR ethics concerns only"]

**Final Justification:**

After reading the rebuttal, I appreciate the author's response and solid work, and all my concerns are well addressed. This work proposed a fast and scalable optimization method for post-training LLMs and significantly boosted the performance and efficiency. As far as I know, it is non-trivial to develop off-policy methods for RL finetuning reasoning models. This work is solid in both algorithm design and experiments. Given these, I increased some sub-scores.

Finally, the studied topic in this paper is really crucial for both academia and industry, considering the huge energy consumption of developing reasoning models.

**Limitations:**

yes, also please include the extra discussion about model predictive task sampling as suggested in the comment.

**Paper Formatting Concerns:**

No concerns

**Quality:**

3

**Strengths And Weaknesses:**

The writing is overall well. I will summarize the merits and flaws of this work based on my understanding.

---

In terms of merits, I think this work is advantageous in four points:

(1) The decoupling desgin is novel. TBA separates data generation from policy updates, leading to efficient resource uses.

(2) The method is scalable and fast.Experiments demonstrated significant wall-clock time improvements, with up to 50× faster training. I think it is extremely beneficial for post-training large models.

(3) TBA can achieve off-policy robustness.The use of trajectory balance enable diverse and scalable off-policy learning.

(4) The experiments are extensive and convincing. It shows strong experimental results across three key LLM tasks with rigorous baseline comparisons.

---

Though there exist no major concerns, I have a few comments as well as suggestions to improve this manuscript. These include:

(1) More discussions on implementation details can be added. Note that the reward model and policy optimization rely on GFlowNets [1] and DPO, and these involve several hyperparameters to tune during post-training.

(2) There requires an efficiency discussion in sequence selection. As mentioned in **Line169-172**, the proposed method generates more samples for each query and selects the subset of generated sequences based on recency and rewards to update the model, this inevitably brings additional memory and inference cost. *When the base model is large enough, e.g., >7B params, the efficiency can be degraded.*

Based on the above analysis, reinforcement finetuning LLMs is expensive in computations, memories and reward annotations. As far as I know, there exist some amortization methods [2-4] to approximately score the sequence reward or predict the preference of tasks in RL. Hence, *I suggest to add a few discussion sentences, i.e., combining amortized evaluation methods in [3,4] with TBA to improve efficiency and robustness in the future* in **Limitation part Line 325-Line 327**.

(3) I want to see the discussion between GRPO [5] in DeepSeek-R1 and asynchronous off-policy methods such as TBA.The analysis is not from quantitative results but some technical discussions. Will removing out the critic be better? I guess this is up to memory constraints.

(4) Some typos, such as Line 277, “these results…” should be “These results”.

**Reference:**

[1] Bengio, Yoshua, et al. "Gflownet foundations." Journal of Machine Learning Research 24.210 (2023): 1-55.

[2] Greenberg, Ido, et al. "Train hard, fight easy: Robust meta reinforcement learning." Advances in Neural Information Processing Systems 36 (2023): 68276-68299.

[3] Wang, Qi Cheems, et al. "Model Predictive Task Sampling for Efficient and Robust Adaptation." (2025).

[4] Qu, Yun, et al. "Fast and Robust: Task Sampling with Posterior and Diversity Synergies for Adaptive Decision-Makers in Randomized Environments." arXiv preprint arXiv:2504.19139 (2025).

[5] Guo, Daya, et al. "Deepseek-r1: Incentivizing reasoning capability in llms via reinforcement learning." arXiv preprint arXiv:2501.12948 (2025).

---

*I would further update my rating of some items if my concerns are well addressed during the rebuttal.*

---

> ### Author Rebuttal · Authors · 2025-07-31
>
> We thank the reviewer for their helpful review, we believe it has improved our submission. Also, we are grateful for the reviewer’s stated interest in updating their ratings based on our responses. We would be happy to address any remaining or new concerns during the discussion period.
>
> > More discussions on implementation details can be added. Note that the reward model and policy optimization rely on GFlowNets [1] and DPO, and these involve several hyperparameters to tune during post-training.
>
> This is a great point! As noted in our response to Reviewer 2z92, our revision will include more details about the reward models used in our experiments, which are from prior work that we emulate the experimental settings of – Noukhovitch et al. (2025) and Lee et al. (2025) for RLHF and red-teaming, respectively. In particular, we will include in our revision the details from their papers on training the reward models that we use. We will also clarify that our GSM8K experiments have programmatically verifiable rewards and thus no reward model.
>
> Regarding policy optimization, many hyperparameter details are in Table 3, Table 4, and Appendix A.3, but our revision will clarify the following missing details. We use AdamW in all experiments with weight decay = 0, except in red-teaming where it is 0.1. For red-teaming, we use learning rate 1e-4 (other experiments have their learning rates provided, e.g. see Table 3). Please let us know if you have any other related concerns that our revision can address!
>
>
> > There requires an efficiency discussion in sequence selection. As mentioned in Line169-172, the proposed method generates more samples for each query and selects the subset of generated sequences based on recency and rewards to update the model, this inevitably brings additional memory and inference cost. When the base model is large enough, e.g., >7B params, the efficiency can be degraded.
> >
> >Based on the above analysis, reinforcement finetuning LLMs is expensive in computations, memories and reward annotations. As far as I know, there exist some amortization methods [2-4] to approximately score the sequence reward or predict the preference of tasks in RL. Hence, I suggest to add a few discussion sentences, i.e., combining amortized evaluation methods in [3,4] with TBA to improve efficiency and robustness in the future in Limitation part Line 325-Line 327… please include the extra discussion about model predictive task sampling as suggested in the comment.
>
> Our revision will address this limitation by including the following discussion, please let us know if you would add anything else! Our TBA implementation samples off-policy data from a buffer without notable overhead by utilizing a maximum buffer size, which we enforce by dropping the oldest rollout data, in order to reduce the time needed for buffer operations. However, future work could explore ways to make TBA’s buffer operations like data selection more efficient for settings that (e.g.) have significantly longer rollouts, must retain all rollout data, need more complex sampling strategies, or have a diversity of tasks [2,3,4].
>
> > I want to see the discussion between GRPO [5] in DeepSeek-R1 and asynchronous off-policy methods such as TBA.The analysis is not from quantitative results but some technical discussions. Will removing out the critic be better? I guess this is up to memory constraints.
>
> Our revision will include the following technical discussion that clarifies the connection between TBA and GRPO. The high-level summary is that these methods have similar update rules and can be similarly performant when data is on-policy, but we emphasize that (as discussed in the DeepSeekMath paper) GRPO is designed for online training, while TBA is designed to work offline or online (consistent with its outperforming GRPO in off-policy settings). Finally, we note that TBA can free up memory on the trainer process by not loading the reference policy onto the trainer process’s GPUs – deviations from the reference policy can instead be computed on the inference process’s GPUs due to TBA’s separation of these processes. Please let us know if you have any questions not answered by the following, more detailed discussion.
>
>
> In the TBA objective, $\log \hat{Z}$ is analogous to the mean of rewards used in GRPO’s advantage computation. Specifically, given a set of model generations for a given query, Equation 4 shows that $\log \hat{Z}$ is an average over two quantities computed for each model generation – one quantity is the generation’s reward and the other quantity reflects the generation’s adherence to the reference policy (the latter quantity is computed as a difference of two terms). In other words, $\log \hat{Z}$ is used to create an advantage-like quantity for each generation that incorporates the deviation from the reference policy, while the deviation from the reference policy (the KL divergence) is separate from the advantage calculation in GRPO.
>
> This is easier to see if you multiply everything inside the square operation in Equation 5 by $-1$, which is equivalent to multiplying the entire equation by $1$ and gives
>
> $\mathcal{L}\_{\text{TB}}^{\text{VarGrad}}(\mathbf{B};\theta) = \frac{1}{BK}\sum_{i=1,j=1}^{i=B,j=K} \bigg(
> \log \pi_{\text{ref}}(\mathbf{y}^{(i,j)} \mid \mathbf{x}^{(i)})- \log \pi_{\theta}(\mathbf{y}^{(i,j)} \mid \mathbf{x}^{(i)})
> \\+ \frac{1}{\beta} r(\mathbf{y}^{(i,j)};\mathbf{x}^{(i)})- \log \hat{Z}(\mathbf{x}^{(i)})  \bigg)^2.$
>
> The gradient with respect to $\theta$ then becomes
>
> $\nabla \mathcal{L}\_{\text{TB}}^{\text{VarGrad}}(\mathbf{B};\theta) = \frac{1}{BK}\sum_{i=1,j=1}^{i=B,j=K} -2 \bigg(
> \log \pi_{\text{ref}}(\mathbf{y}^{(i,j)} \mid \mathbf{x}^{(i)}) - \log \pi_{\theta}(\mathbf{y}^{(i,j)} \mid \mathbf{x}^{(i)})
> \\+ \frac{1}{\beta} r(\mathbf{y}^{(i,j)};\mathbf{x}^{(i)}) - \log \hat{Z}(\mathbf{x}^{(i)})  \bigg)\nabla \log \pi_{\theta}(\mathbf{y}^{(i,j)} \mid \mathbf{x}^{(i)}),$
>
> and we can turn this into a maximization problem instead of a minimization problem by multiplying the right side by $-1$, demonstrating that $\log \hat{Z}$ does indeed act similarly to GRPO’s mean of rewards inside an advantage-like term that weights the gradient of the policy:
>
> $\nabla \mathcal{J}\_{\text{TB}}^{\text{VarGrad}}(\mathbf{B};\theta) = \frac{1}{BK}\sum_{i=1,j=1}^{i=B,j=K} 2 \bigg(
> \log \pi_{\text{ref}}(\mathbf{y}^{(i,j)} \mid \mathbf{x}^{(i)}) - \log \pi_{\theta}(\mathbf{y}^{(i,j)} \mid \mathbf{x}^{(i)})
> \\+ \frac{1}{\beta} r(\mathbf{y}^{(i,j)};\mathbf{x}^{(i)}) - \log \hat{Z}(\mathbf{x}^{(i)})  \bigg)\nabla \log \pi_{\theta}(\mathbf{y}^{(i,j)} \mid \mathbf{x}^{(i)}).$
>
> > Some typos, such as Line 277, “these results…” should be “These results”.
>
> Thank you for noting this! Our revised manuscript is being thoroughly proofread to address this typo and a few others that we found (e.g., the subscript $\phi$ was previously missing from the reward function in Equation 5 and has been added to the revision).
>
> [1] Bengio, Yoshua, et al. "Gflownet foundations." Journal of Machine Learning Research 24.210 (2023): 1-55.
>
> [2] Greenberg, Ido, et al. "Train hard, fight easy: Robust meta reinforcement learning." Advances in Neural Information Processing Systems 36 (2023): 68276-68299.
>
> [3] Wang, Qi Cheems, et al. "Model Predictive Task Sampling for Efficient and Robust Adaptation." (2025).
>
> [4] Qu, Yun, et al. "Fast and Robust: Task Sampling with Posterior and Diversity Synergies for Adaptive Decision-Makers in Randomized Environments." arXiv preprint arXiv:2504.19139 (2025).
>
> [5] Guo, Daya, et al. "Deepseek-r1: Incentivizing reasoning capability in llms via reinforcement learning." arXiv preprint arXiv:2501.12948 (2025).

---

> ### Comment · Reviewer_ZkfV · 2025-08-02
> **Good Paper**
>
> **After reading the rebuttal, I appreciate the author's response and solid work, and all my concerns are well addressed.** This work proposed a fast and scalable optimization method for post-training LLMs and significantly boosted the performance and efficiency. As far as I know, it is non-trivial to develop off-policy methods for RL finetuning reasoning models. This work is solid in both algorithm design and experiments. Given these, I increased some sub-scores.
>
> More responses to the authors' rebuttal are as follows.
>
> (1) Glad to see the added MPTS series discussions in Line 325-Line 327 of the paper.
>
> As the field develops so fast, some sampling efficiency issue gradually emerges, and combining the proposed TBA with some method like the MPTS series, e.g., MoPPS (model predictive prompt selection) , for active task selection is indeed promising to further cut off required computations as future work. These new discussion contents complement the limitations and future work part.
>
> (2)Analysis on GRPO is clear and sufficient. TBA provides a new strategy to achieve GRPO's purpose and retains its advantage. It is also great to include these discussions for readers to have better understanding.
>
> **Finally, the studied topic in this paper is really crucial for both academia and industry, considering the huge energy consumption of developing reasoning models.**

---

> > ### Author Response · Authors · 2025-08-05
> >
> > We are grateful for your support and suggestions for the revision.
> >
> > Thank you!
> >
> > Authors of 11130

---

### Official Review · Reviewer_2z92 · 2025-07-02

**Clarity:** 3
**Significance:** 3
**Originality:** 3
**Rating:** 4
**Confidence:** 4

**Summary:**

This paper proposes a method called TBA (Trajectory Balance with Asynchrony) to improve the post-training process of large language models (LLMs). The approach combines the Trajectory Balance (TB) objective with an asynchronous reinforcement learning (RL) framework, aiming to enhance training efficiency, diversity, and scalability in sparse-reward scenarios. Experiments on mathematical reasoning (GSM8K), preference fine-tuning (PFT), and automated red-teaming (RT) tasks demonstrate TBA's advantages in speed and performance over baseline methods (e.g., PPO, RLOO, Online DPO). The authors emphasize the method's open-source availability and reproducibility while discussing its broader societal impact and limitations.

**Questions:**

Based on the concerns raised in the weaknesses section, please provide explanations and supplementary experiments addressing the following:

1. Theoretical Justification of TB + Asynchronous Combination:
   - Can you formally derive the relationship between TB and other RL objectives (e.g., PPO)?
   - Why does combining TB with asynchronous search yield better performance than using TB alone in a synchronous setting?

2. Variance Analysis in TB Estimation:
   - Can you provide theoretical bounds on the gradient variance of the TB objective under different sequence lengths?
   - What happens when $K$ (number of responses per prompt) is increased further—does performance continue to improve?

3. Convergence Behavior Under Delayed Policies:
   - Does TBA converge under the assumption of a delayed policy $\pi_{\theta'}$?
   - Can you implement a theoretically grounded delay correction mechanism (e.g., importance sampling, gradient compensation) and compare it with your current heuristic approach?

4. Robustness Against Adversarial or Noisy Rewards:
   - How does TBA perform when the reward model is perturbed or when off-policy data includes adversarial examples?
   - Does reward-based prioritization amplify the effect of noisy or misleading rewards?

5. Fair Comparison with Strong Off-Policy Baselines:
   - Please include comparisons with RAFT [9], asynchronous PPO [40], and other advanced off-policy methods.
   - Are the claimed performance gains robust under more comprehensive evaluations?

**Ethical Concerns:**

["NO or VERY MINOR ethics concerns only"]

**Final Justification:**

The author's response has addressed my concerns, and I keep my initial rating of Borderline Accept.

**Limitations:**

Yes.

**Quality:**

3

**Strengths And Weaknesses:**

Strengths:
--TBA effectively combines trajectory balance (a physics-inspired equilibrium condition) with asynchronous RL, addressing bottlenecks in training efficiency and sample diversity for LLM post-training. This integration shows clear value, particularly in sparse-reward settings.
--Rigorous evaluation across multiple tasks (mathematical reasoning, preference tuning, red-teaming) with comparisons against strong baselines (PPO, RLOO, Online DPO).
--Ablation studies analyze key hyperparameters (e.g., KL coefficient β, batch size), supported by visualizations with error bars, enhancing credibility.
--Code and experimental details (Appendix A) are publicly shared, including hardware specs (GPU models/counts) and training protocols, aligning with NeurIPS reproducibility standards.
--Clear reporting of computational costs (e.g., runtime per experiment, GPU memory usage) aids future research.
--Section 6 candidly discusses limitations (e.g., sensitivity to low-resolution inputs, high compute costs) and misuse risks (e.g., disinformation generation), proposing mitigation strategies (e.g., controlled model access). This aligns with academic ethics.

Weaknesses:
-- Lack of Theoretical Innovation in Combining Existing Techniques: The core components of TBA—the trajectory balance objective and the asynchronous distributed architecture—are both existing techniques (from GFlowNets and IMPALA, respectively). While the paper is the first to combine them in the context of LLM post-training, it does not sufficiently explain why this combination can overcome the limitations of existing off-policy training approaches (e.g., [40] highlights that off-policyness harms performance). The theoretical novelty is weak. Moreover, the lack of comparison with recent high-performance off-policy methods (e.g., RAFT [9]) undermines the competitiveness of TBA’s central claim of "efficient use of replay buffers."

--Empirical Variance Control May Conceal TB Objective's Theoretical Deficiencies: The TB objective relies on VarGrad estimation $\log \hat{Z}$ (Formula 4), but the paper does not prove the bias bounds of this estimator in long-sequence generation. For instance, when sequence length = 512, the cumulative error of $\log \pi_{\theta}$ may grow super-linearly. Appendix B shows that GSM8K training diverges when the KL coefficient $\beta < 0.003$ (Figure 6, bottom left), suggesting high variance issues. However, the authors only address this problem empirically—by increasing the number of samples ($K=20$) and raising $\beta$. Thus, an important question arises: Is TBA’s performance gain derived from engineering patches to mitigate TB’s flaws, rather than algorithmic superiority? Would these empirical strategies fail in more complex tasks (e.g., long-text reasoning)?

--Does Asynchronicity Undermine TB’s Convergence Guarantees?  TB theoretically requires data distribution stationarity ([4]), but in TBA, the SEARCHER uses a delayed policy $\pi_{\theta'}$ to generate data, while the TRAINER updates $\pi_{\theta}$, leading to non-stationary data distributions. The paper employs heuristic methods (sync interval $k$, recency sampling $m$) to manage off-policyness, but experiments show that when $m=0.4$, the Win Rate on TL;DR drops significantly (0.67 vs. 0.8), indicating residual asynchronous bias. Why not introduce theoretically grounded delay correction mechanisms? Is the current approach essentially masking convergence risks through hyperparameter tuning?

--Reward Model Vulnerability Threatens TB’s Robustness:The TB objective directly optimizes $\pi_{\theta} \propto \pi_{\text{ref}} \exp(\beta^{-1} r_{\phi})$, but reward models are known to be vulnerable to adversarial examples (e.g., reward hacking [55]). The paper does not test how TBA performs when off-policy data contains adversarial samples (e.g., intentionally constructed high-reward yet meaningless responses). Furthermore, TBA’s reward-based prioritization strategy (Section 4.2) may amplify the influence of such samples. In real-world scenarios where reward models are imperfect, is TB more fragile compared to more constrained RLHF methods like DPO? Why does the paper avoid validating this risk?

--Experimental Design May Systematically Underestimate Baseline Methods: The paper does not compare with contemporary high-performance off-policy methods (e.g., RAFT [9] with ranking-based fine-tuning) or modified asynchronous frameworks (e.g., asynchronous PPO from [40]). Do the claimed “significant advantages” of TBA stem from insufficient baseline comparisons or biased evaluation metrics? If fairly compared, would TBA still maintain its overwhelming superiority?

---

> ### Author Rebuttal · Authors · 2025-07-31
>
> Thank you! This review improved our submission, and we would be happy to address any remaining or new concerns.
>
> # Theoretical Justification of TB + Asynchronous Combination
>
> > Can you formally derive the relationship between TB and other RL objectives (e.g., PPO)?
>
> Our response to reviewer C5mN derives the relationship between the gradients of TBA and the off-policy robust Proximal RLOO, and it provides intuition for the two approaches’ strategies to dealing with off-policy data, which we will provide in our revision.
>
> More broadly, the gradient of the TB loss when evaluated with on-policy samples is exactly equal to the REINFORCE gradient with a mean baseline (equivalent to RLOO gradient within constant scaling factor), evaluated with the KL regularized reward where beta (the temperature) is the KL coefficient (Malkin et al., 2023). Our submission (Line 109) notes that TB’s gradient is equivalent to the gradient of reverse KL between the training policy and the target density $p\_{ref}(x)exp(\frac{1}{\beta}r(x))$, and our revision will illustrate this mathematically as well.
>
> Finally, our revision will note that, in the off-policy setting, TB does not have any precise equivalence to other on-policy RL objectives, but for LLMs is still equivalent to PCL (path consistency learning, Nachum et al., 2017).
>
> Regarding RAFT, for binary rewards, on-policy RAFT is exactly REINFORCE without a baseline. Off-policy RAFT is REINFORCE without a baseline, and without off-policy importance weighting. Accordingly, we believe the baselines we compare against are sufficient, especially since we include off-policy REINFORCE/RLOO.
>
> > Why does combining TB with asynchronous search yield better performance than using TB alone in a synchronous setting?
>
> While it’s true that TBA (TB with asynchronous search) leads to better training speed than TB in a synchronous setting, our submission did not find that there were accuracy/win-rate benefits of asynchrony. Specifically, TBA leads to speed ups because the trainer process does not pause for generations as it does in synchronous settings, and we found that moving closer to a synchronous setting does provide expected accuracy benefits, especially for GSM8K with RhoMath-1B – e.g., see Figure 6’s ablation studies that reduce the sync interval $k$ (top right plot) and increase the probability of sampling recent generations $m$ (top left plot).
>
> # Variance Analysis in TB Estimation
>
> > [theoretical bounds]?
> >
> > What happens when $\mathbf{K}$ (number of responses per prompt) is increased further—does performance continue to improve?
>
> Doubling $\mathbf{K}$ leads to a slight performance improvement and variance reduction, as shown in the following results that we will include in a new ablation study in our revision’s appendix.
>
> **Table R1. RL finetuning of RhoMath-1B on GSM8K data.**
> | Method | Test Accuracy | Standard Deviation |
> |--------|--------|------|
> | TBA, K=20 | 54.61  | 0.85 |
> | TBA, K=40 | 54.89  | 0.49 |
>
> **Critically, addressing this limitation by increasing $\mathbf{K}$ does not come at any compute cost.** Specifically, when increasing $\mathbf{K}$, we do not increase the total number of rollouts, we simply reduce the number of unique queries per batch as discussed in Footnote 1 of our submission (Line 254), generating more rollouts per query and maintaining batch size. Moreover, both of these accuracies are significantly beyond the accuracies achieved by prior work that uses the same setup and significantly longer training times.
>
> Our revision will include theoretical bounds on the gradient variance as a future direction.
>
>
> # Convergence Behavior Under Delayed Policies
>
> > Does TBA converge under the assumption of a delayed policy $\pi_{\theta’}$?
> >
> > Can you implement a theoretically grounded delay correction mechanism (e.g., importance sampling, gradient compensation) and compare it with your current heuristic approach?
>
> Yes. We would like to clarify that the only requirements for the convergence of TB is taking the loss for all complete trajectories to 0 [4,34]. Crucially, this guarantee does not depend on the policy generating the trajectories, which is what makes the objective suitable for off-policy training, as noted in [4].
>
> Note that this convergence does not rely on any external mechanism like importance sampling, etc. Moreover, we wish to clarify that our experience is that TBA does not require hyperparameter tuning beyond what is required for other RL methods like PPO, GRPO, etc. Relatedly, the reviewer correctly states that RLHF ablation studies in Section 5.4 show that win-rates are finally harmed when $m$ falls to $0.4$, and the GSM8K ablation studies similarly show how accuracy can decline with $m$ (Figure 6). However, we emphasize that these results are not necessarily due to any flaw in TB or its off-policy robustness; rather, these results may relate to the fact that increasing the off-policyness can nearly eliminate the amount of feedback the rollout policy receives when training LLMs for a fixed step count to facilitate rapid training (about 1 hour on 4 GPUs).
>
> # Robustness Against Adversarial or Noisy Rewards
>
> > How does TBA perform when the reward model is perturbed or when off-policy data includes adversarial examples?
> >
> > Does reward-based prioritization amplify the effect of noisy or misleading rewards?
>
> We plan to incorporate the following discussion in the revised manuscript to address this important matter.
>
> In our RLHF experiments, we perform test-data evaluations with a large ‘gold-standard’ reward model, and we perform training with a smaller proxy reward model that was trained on rankings from the gold-standard model (i.e., we follow the setup of [40]). Thus, our setup provides an opportunity to determine the potential for overoptimizing feedback from an imperfect reward model. This is important to test because TBA’s reward-based prioritization could, through identifying and selecting higher-reward samples, accelerate reward hacking. Potentially offsetting this, however, TBA operates asynchronously on off-policy data, which may slow reward hacking by increasing the time it takes for the searcher policy to be updated with information about high-reward samples. Additionally, like other methods, TBA may reduce reward hacking by penalizing deviation from the reference policy via $\beta$.
>
> We test whether overoptimization of the imperfect proxy model occurs by simultaneously considering win-rate according to the ‘gold-standard’ reward model and the divergence from the reference policy via evaluation of the reference policy’s perplexity on the RLHF policy’s generations. Again, this setup follows [40], which notes that the latter metric can be thought of as a proxy for the KL divergence from the reference policy. If the latter metric is high (i.e., the reference policy is surprised by the generations) or the former metric is low (i.e., the gold-standard reward model does not grade the generations highly), then there is evidence for overoptimization of the proxy model. Critically, Figure 4 shows that TBA establishes a new Pareto frontier for these two metrics, surpassing that of DPO’s, which had previously been shown to outperform PPO and RLOO on these metrics [40].
>
> Therefore, our findings are consistent with TBA discouraging reward hacking relative to other post-training methods. Like many methods, TBA may discourage reward hacking through the $\beta$ parameter, but it also may avoid reward hacking via its asynchronous training on off-policy data, which increases the number of optimization steps between the searcher policy’s generation of a rollout and the searcher policy’s parameter update based on that rollout. In other words, TBA prevents the searcher policy from receiving frequent feedback about the value of the samples it generates, which may explain why (relative to competing methods) it did not overoptimize imperfect proxy models in our experiments.
>
> # Fair Comparison with Strong Off-Policy Baselines
>
> > Please include comparisons with RAFT [9], asynchronous PPO [40], and other advanced off-policy methods. Are the claimed performance gains robust under more comprehensive evaluations?
>
> Addressing this concern, the revision will provide the following comparison to the advanced off-policy methods Online DPO and Proximal RLOO, as well as an async PPO baseline that is robust to moderately off-policy data [40]. The revision will also include a theoretical comparison of the gradients of TBA and the off-policy robust Proximal RLOO, which we provide in our response to Reviewer C5mN.
>
> **Table R2. RL finetuning of Pythia 410M on TL;DR data (Compute = 4xA100s).**
> | Method | Perplexity ↓ | Win Rate ↑ | Training Steps Before Syncing |
> |--------|----------------|------------|------------------|
> | Async PPO	     | 1.14       	| 0.84   	| 2          	|
> | Proximal RLOO | 1.13       	| 0.83   	| 2          	|
> | Online DPO	     | 1.13       	| 0.85  	| 2          	|
> | TBA (ours)         | 1.13       	| 0.86   	| 10        	|
>
> Critically, TBA outperforms three advanced off-policy methods, achieving both the highest win rate and the lowest perplexity – i.e., the closest adherence to the reference policy, consistent with low reward-hacking. TBA also makes training significantly faster, compared to baselines, and trains with stability on data that is relatively more off-policy (10 steps instead of 2).
>
> Beyond off-policy method comparisons, we emphasize that TBA can outperform state-of-the-art on-policy methods learning synchronously: Figures 1 and 3 show that TBA outperforms *synchronous on-policy* Online DPO, while Table 1 shows that TBA outperforms *synchronous on-policy* RLOO and PPO. Moreover, in the context of Table R2 above, Synchronous PPO with on-policy data obtains the same win-rate as TBA (0.86). Thus, TBA leverages off-policy learning to obtain the speed benefits of asynchrony while matching or surpassing the accuracy/win-rate of on-policy learning.

---

> > ### Author Response · Authors · 2025-08-06
> >
> > Hello Reviewer 2z92,
> >
> > Thanks again for your review. We’ve posted a comprehensive rebuttal that we believe addresses all of the points you raised. If you have additional questions or things to discuss, please let us know. If we have addressed all of your concerns, we’d appreciate your according support of our submission's acceptance.
> >
> > Thank you,
> >
> > Authors of 11130

---

### Official Review · Reviewer_C5mN · 2025-07-07

**Clarity:** 2
**Significance:** 3
**Originality:** 3
**Rating:** 3
**Confidence:** 3

**Summary:**

This paper proposes a “Trajectory Balance with Asynchrony (TBA)” distributed reinforcement learning framework to improve the post-training efficiency and performance of LLMs. TBA achieves asynchronous training by decoupling data generation and policy updates. It leverages the offline policy training objective based on Trajectory Balance (TB) and utilizes large-scale computing on search, which enhances exploration and data diversity. Experimental results on multiple tasks show that this method has faster training speed and comparable performance compared to existing methods.

**Questions:**

1.	As mentioned in Weaknesses 3, does this method show performance and efficiency improvement when computing resources are limited in practice?

2.	In writing, abbreviations that appear for the first time should be written in full, such as GFlowNets in the abstract. What exactly does MCMC refer to in line 107? Please explain and use the full name to avoid potential ambiguity.

**Ethical Concerns:**

["NO or VERY MINOR ethics concerns only"]

**Limitations:**

yes

**Quality:**

3

**Strengths And Weaknesses:**

Strengths:
1.	This work decouples data generation and strategy updates to achieve asynchronous operations, which improves training efficiency.

2.	TBA increases the diversity of data in the experience replay buffer by utilizing a large-scale offline strategy sampling.

3.	In three common LLM post-training RL pipelines, TBA demonstrates performance comparable to or better than existing baselines, while training faster.

Weaknesses:
1.	In the Red-teaming, the used base model GPT-2 is a bit outdated. Many more advanced models have been released. Moreover, the base LLM sizes used for the three pipelines all include 1B, such as RhoMath-1B and Pythia 1B, with the smallest being 410M and the largest being 8B. The effectiveness and sensitivity of this method on larger sizes of LLMs have not been explored.

2.	The motivation for this method seems insufficient. The motivation for proposing the off-policy training objective in this paper stems from the fact that existing RL algorithms, such as PPO and RLOO, suffer from on-policy constraints, but there is a lack of analysis and discussion compared with existing off-policy methods. Therefore, a more comprehensive analysis and comparison with off-policy-based methods are needed to reinforce the motivation behind the proposed method.

3.	The proposed method enables scalable search with large-scale computing resources, which contributes to the effectiveness of the method. So, does this method still have advantages when computing resources are limited in practice?

4.	The writing needs improvement. The section 3 preliminaries, which typically introduces prior knowledge, takes up one and a half pages, while the section 4, the writing of the core proposed method TBA, only takes one page. Figure 1 shows the experimental results, which are referred to in the experiments of Section 5, but why is this figure placed below the abstract on the first page?

---

> ### Author Rebuttal · Authors · 2025-07-31
>
> We thank the reviewer for their thorough review, we believe that our addressing their questions has notably improved the manuscript. We understand that the reviewer’s main concerns are wanting to see TBA tested with a larger model, TBA’s connections to other methods, and clarity issues. We would be happy to know that the reviewer finds their concerns addressed and supports our submission’s acceptance, and we are equally interested in discussing any remaining or new concerns.
>
> > .... GPT-2 is a bit outdated. ... base LLM sizes used for the three pipelines all include 1B, ... The effectiveness and sensitivity of this method on larger sizes of LLMs have not been explored.
>
> Notably, our submission’s red-teaming section contained experiments with GPT-2 and Llama 3 (1B and 8B). However, we agree that GPT-2 is “a bit outdated” and have added the following discussion to our revision to clarify why we consider GPT-2: We study GPT-2 for the following reasons:
> - prior work published in ICLR 2025 on automated red-teaming utilized GPT-2 as a baseline, and we adopt it here to facilitate comparisons to Lee et al., 2025;
> - it is small enough to fit onto V100 GPUs, which allows us to test the effects of scaling TBA’s number of searcher/trainer processes to 64 when using 64 V100 GPUs; and
> - models like GPT2 are particularly well-suited for automated red-teaming since they haven’t been explicitly trained on filtered datasets or safety-tuned, resulting in more effective red-teaming prompts.
>
> Also, we would like to note that our RLHF experiments consider larger 2.8B parameter models, where we found TBA effective. However, we do agree that testing even larger models would make the results stronger. Considering the short duration of the rebuttal period, we considered a preliminary experiment on the MATH dataset with the Qwen2.5-7B-Instruct model. We use the MATH dataset for training and evaluate the performance on MATH-500. We compare asynchronous GRPO to TBA, finding the latter excels. We note that these are preliminary results, and we will discuss these in the next revision.
>
> **Table R1. RL tuning Qwen2.5-7B-Instruct on MATH**
> | Method | MATH-500 Accuracy |
> |-|-|
> | TBA | 70% |
> | Async GRPO | 67.3% |
>
> > ... there is a lack of analysis and discussion compared with existing off-policy methods. Therefore, a more comprehensive analysis and comparison with off-policy-based methods are needed to reinforce the motivation behind the proposed method.
>
> Below, we first clarify that we provided empirical comparisons to strong off-policy approaches (and on-policy approaches). Addressing the reviewer’s concerns, we then show how we improve our revision with new empirical comparisons to an off-policy version of RLOO (Proximal RLOO) [40] as well as asynchronous PPO. Finally, we discuss how our revision provides a new comparison of the gradients of Proximal RLOO and TBA, highlighting key differences in these off-policy robust approaches.
>
> First, to clarify our submission, we note that our experiments extensively compared TBA to a strong off-policy method, DPO, which we gave benchmark values for in both synchronous and asynchronous (off-policy) settings. We also compared TBA to on-policy methods operating in synchronous, on-policy settings (Table 1). TBA exclusively used off-policy data in these experiments, training rapidly in our asynchronous framework. However, TBA also produced the highest win-rates/accuracies in each comparison, demonstrating comprehensive improvements.
>
> Second, we provide the following comparison to the off-policy methods Online DPO and Proximal RLOO, as well as an async PPO baseline that is robust to moderately off-policy data [40]. Critically, TBA outperforms all of these methods, achieving both the highest win rate and the lowest perplexity (i.e., the closest adherence to the reference policy). TBA also makes training significantly faster, compared to baselines, and trains with stability on data that is relatively more off-policy (10 steps instead of 2).
>
> **Table R2. RL finetuning of Pythia 410M on TL;DR data (Compute = 4xA100s).**
> | Method | Perplexity ↓ | Win Rate ↑ | Training Steps Before Syncing |
> |--------|----------------|------------|------------------|
> | Async PPO	     | 1.14       	| 0.84   	| 2          	|
> | Proximal RLOO | 1.13       	| 0.83   	| 2          	|
> | Online DPO	     | 1.13       	| 0.85  	| 2          	|
> | TBA (ours)         | 1.13       	| 0.86   	| 10        	|
>
> Finally, we use gradient analysis to compare TBA to Proximal RLOO, an off-policy variant of RLOO with empirically demonstrated off-policy robustness [40]. With k=3 samples, the gradient for Proximal RLOO for a single sample $y_1$ is
>
> $\nabla \mathcal{J}\_{\text{P-RLOO}}(\theta) = \frac{\pi_\theta(y_1|x)}{\pi_{ref}(y_1|x)}  \hat{A}(y_1|x) \nabla_\theta \log \pi_\theta(y_1|x),$
>
> where the advantage $\hat{A}(y_1|x) = R(y_1, x) - \frac{1}{k-1} \sum_{i \neq 1} R(y_i, x)$.
>
> The gradient for TBA is
>
>  $\nabla \mathcal{J}\_{\text{TB}}(\theta) = 2 \bigg( \log \pi_{\text{ref}}(y_1 \mid x) - \log \pi_{\theta}(y_1 \mid x) + \frac{1}{\beta} r(y_1; x) - \log \hat{Z}(x)  \bigg)\nabla \log \pi_{\theta}(y_1 \mid x),$
> where $\log \hat{Z}(x)= \frac{1}{k} \sum_{i}  \bigg( \log \pi_{\text{ref}}(y_i \mid x) - \log \pi_{\theta}(y_i \mid x) + \frac{1}{\beta} r(y_i; x) \bigg).$
>
> We emphasize that $\log \hat{Z}(x)$ plays a role similar to the mean of rewards used to compute the advantage in the Proximal RLOO gradient. The main differences in these methods are: (1) TBA's $\log \hat{Z}(x)$ average includes both reward and deviations from the reference policy, normalizing the latter when Proximal RLOO does not. (2) The term that precedes the gradient of the policy in TBA is increasing as the sampled sequence $y_1$ becomes more probable under the reference policy, whereas the analogous term in Proximal RLOO $\frac{\pi_\theta(y_1|x)}{\pi_{ref}(y_1|x)}$ decreases as $y_1$ becomes more probable under the reference policy. In summary, Proximal RLOO seeks to deal with off-policy data by up-weighting gradients on samples that are relatively probable under the current policy; in contrast, TBA manages off-policy data by balancing rewards with deviations from the reference policy, up-weighting gradients on samples with higher probability under the reference policy (all else equal).
>
> > The writing needs improvement. The section 3 preliminaries, which typically introduces prior knowledge, takes up one and a half pages, while the section 4, the writing of the core proposed method TBA, only takes one page. Figure 1 shows the experimental results, which are referred to in the experiments of Section 5, but why is this figure placed below the abstract on the first page?
>
>
> Regarding the Figure 1 placement, we included it below the abstract to give readers an immediate depiction of the efficiency improvements TBA creates, but we see the clarifying benefit of having it in the experiments section and will happily move it there if that would improve your rating of our submission and its clarity.
>
> Regarding the preliminary section’s length, we note that our submission’s preliminaries actually occupy fewer lines (Line 93 – Line 139) than the description of our method TBA (Line 141 – Line 191), but this was not clear in our submission because the visual illustration of TBA in Figure 2 appeared in the same section as the preliminaries – our revision will fix this Figure 2 placement such that description of our core proposed method occupies about 1.5 pages, and the preliminaries section occupies about 1 page. We would be happy to shorten this preliminaries section further if you recommend that, but we note that it does include background on and motivation for the loss function we use (Equation 5), which we believe adds clarity.
>
>
> > The proposed method enables scalable search with large-scale computing resources, which contributes to the effectiveness of the method. So, does this method still have advantages when computing resources are limited in practice?... As mentioned in Weaknesses 3, does this method show performance and efficiency improvement when computing resources are limited in practice?
>
> Our submission demonstrated that, yes, our proposed TBA approach produces improved “performance and efficiency” when computing resources are limited in practice. We have revised our submission to make this more clear. In particular, our original submission discussed both “compute-matched” experiments (e.g., Line 267) and “resource scaling” experiments (e.g., Line 308), and we have revised these discussions to specifically mention the number of GPUs involved in each type of experiment.
>
> For example, instead of only mentioning that our compute-matched experiments use 4 GPUs in the captions of associated figures (Figure 1 and Figure 3), we now note this usage of only 4 GPUs in the text discussing these results. Use of 4 GPUs is standard in these RL experiments (e.g. it was adopted by the baseline methods we compare to). We also go beyond the baselines we compare to, using up to 64 GPUs, in our resource-scaling experiments – our submission only described these experiments’ GPU counts in Table 2 and Appendix A, and our revision will clarify these resource-scaling experiments’ GPU counts in the writing of the main text as well.
>
> > In writing, abbreviations that appear for the first time should be written in full, such as GFlowNets in the abstract. What exactly does MCMC refer to in line 107? Please explain and use the full name to avoid potential ambiguity.
>
> Thank you for noting these potential ambiguities, our revision will address both of the mentioned examples as well as others – e.g., we will write the full names of all baseline methods like “Group Relative Policy Optimization” for GRPO. Regarding MCMC, our revision clarifies that this refers to “Markov chain Monte Carlo”, a type of sampling strategy that is too slow to serve as a replacement for post-training LLMs in most settings.

---

> > ### Author Response · Authors · 2025-08-06
> >
> > Hello Reviewer C5mN,
> >
> > Thanks again for your review. We’ve posted a comprehensive rebuttal that we believe addresses all of the points you raised. If you have additional questions or things to discuss, please let us know. If we have addressed all of your concerns, we’d appreciate your according support of our submission's acceptance.
> >
> > Thank you,
> >
> > Authors of 11130

---

### Official Review · Reviewer_cZos · 2025-07-09

**Clarity:** 3
**Significance:** 3
**Originality:** 3
**Rating:** 5
**Confidence:** 4

**Summary:**

The paper proposes an interesting method to conduct LLM post-training without using policy-gradient style approach. The method is about 4x faster than GRPO (Figure 1's 50x faster claim is massively misleading). The domain is on safety training, which is a very important domain for LLM post-training.

**Questions:**

I do have a few questions about the method, since I'm only familiar with traditional RL methods.

The equation 5, if I actually expand $\log \hat Z$, then the equation becomes a square loss term to train the reward function: $r_{\phi}(y, x) - r(y, x)$, but then all the policy $\pi$ related terms will get cancelled out.

I'm actually not sure how the training is done -- how is $\theta$ learned? This is probably because I'm not familiar with the Bayesian notation, but understanding this will help me understand the main difference between this objective and GRPO style objective. I'm happy to raise my score after the discussion round!

**Ethical Concerns:**

["NO or VERY MINOR ethics concerns only"]

**Final Justification:**

The rebuttal is excellent and has addressed all my concerns.

**Limitations:**

Yes

**Quality:**

3

**Strengths And Weaknesses:**

Strengths:
1. The paper is well-written. The method seems very clear.
2. The benefits of async RL are well-documented.
3. Using GFlowNet for RL seems novel (but frankly, I'm not familiar enough with the Bayesian RL literature)
4. The implementation seems easy enough with existing libraries, which is a huge plus.

Weaknesses:
1. GFlowNet is already an established method. The paper does not offer innovation on methodology beyond the existing method. However, applying it for LLM post-training is interesting, and the empirical explorations seem warranted.

---

> ### Author Rebuttal · Authors · 2025-07-31
>
> We thank the reviewer for their detailed and thorough review, and we are grateful for their stated interest in raising their score after the discussion round! Their efforts have helped us improve the revision, and we would be happy to discuss any additional or remaining concerns.
>
> > The method is about 4x faster than GRPO (Figure 1's 50x faster claim is massively misleading).
>
> We will improve the specificity of our text to avoid misleading. While our results in Figure 1 demonstrate that TBA’s speedups can be as much as 50x, it’s correct that we only find speedups this large when comparing to methods that conduct additional rollouts to estimate the value of candidate states (i.e. VinePPO). Our text description of our results will clarify that the speedups relative to common methods like GRPO are approximately 4x, while the speedup relative to more complex methods like VinePPO that get closer to TBA’s elevated accuracy are 50x.
>
> > The equation 5, if I actually expand $\log \hat{Z}$, then the equation becomes a square loss term to train the reward function: $r_{\phi}(\mathbf{y};\mathbf{x}) - r(\mathbf{y};\mathbf{x})$, but then all the policy related terms will get cancelled out… I'm actually not sure how the training is done -- how is $\theta$ learned? This is probably because I'm not familiar with the Bayesian notation, but understanding this will help me understand the main difference between this objective and GRPO style objective. I'm happy to raise my score after the discussion round!
>
> Great question! In our objective, $\log \hat{Z}$ is analogous to the mean of rewards used in GRPO’s advantage computation. Specifically, given a set of model generations for a given query, Equation 4 shows that $\log \hat{Z}$ is an average over two quantities computed for each model generation – one quantity is the generation’s reward and the other quantity reflects the generation’s adherence to the reference policy (the latter quantity is computed as a difference of two terms). Since $\log \hat{Z}$ is an average over multiple generations, including $\log \hat{Z}$ in Equation 5 does not cause cancellation. Instead, $\log \hat{Z}$ is used to compute per-generation deviations from the mean, creating an advantage-like quantity for each generation.
>
> This is easier to see if you multiply everything inside the square operation in Equation 5 by $-1$, which is equivalent to multiplying the entire equation by $1$ and gives
>
> $\mathcal{L}\_{\text{TB}}^{\text{VarGrad}}(\mathbf{B};\theta) = \frac{1}{BK}\sum_{i=1,j=1}^{i=B,j=K} \bigg(
> \log \pi_{\text{ref}}(\mathbf{y}^{(i,j)} \mid \mathbf{x}^{(i)})- \log \pi_{\theta}(\mathbf{y}^{(i,j)} \mid \mathbf{x}^{(i)})+ \frac{1}{\beta} r(\mathbf{y}^{(i,j)};\mathbf{x}^{(i)})- \log \hat{Z}(\mathbf{x}^{(i)})  \bigg)^2.$
>
> The gradient with respect to $\theta$ then becomes
>
> $\nabla \mathcal{L}\_{\text{TB}}^{\text{VarGrad}}(\mathbf{B};\theta) = \frac{1}{BK}\sum_{i=1,j=1}^{i=B,j=K} -2 \bigg(
> \log \pi_{\text{ref}}(\mathbf{y}^{(i,j)} \mid \mathbf{x}^{(i)}) - \log \pi_{\theta}(\mathbf{y}^{(i,j)} \mid \mathbf{x}^{(i)})+ \frac{1}{\beta} r(\mathbf{y}^{(i,j)};\mathbf{x}^{(i)}) - \log \hat{Z}(\mathbf{x}^{(i)})  \bigg)\nabla \log \pi_{\theta}(\mathbf{y}^{(i,j)} \mid \mathbf{x}^{(i)}),$
>
> and we can turn this into a maximization problem instead of a minimization problem by multiplying the right side by $-1$, demonstrating that $\log \hat{Z}$ does indeed act similarly to GRPO’s mean of rewards inside an advantage-like term that weights the gradient of the policy:
>
> $\nabla \mathcal{J}\_{\text{TB}}^{\text{VarGrad}}(\mathbf{B};\theta) = \frac{1}{BK}\sum_{i=1,j=1}^{i=B,j=K} 2 \bigg(
> \log \pi_{\text{ref}}(\mathbf{y}^{(i,j)} \mid \mathbf{x}^{(i)}) - \log \pi_{\theta}(\mathbf{y}^{(i,j)} \mid \mathbf{x}^{(i)})+ \frac{1}{\beta} r(\mathbf{y}^{(i,j)};\mathbf{x}^{(i)}) - \log \hat{Z}(\mathbf{x}^{(i)})  \bigg)\nabla \log \pi_{\theta}(\mathbf{y}^{(i,j)} \mid \mathbf{x}^{(i)}).$
>
> Importantly, even though $\log \hat{Z}$  is also a function of $\theta$, the gradient above does not go through $\log \hat{Z}$ because  $\log \hat{Z}$ is detached from the compute graph.  Our submission had mentioned that $\log \hat{Z}$ is detached on Line 134, but we will emphasize this further in our revision by adding the following stop-grad notation to Equation 5: $\text{STOP-GRAD}(\log \hat{Z}(\mathbf{x}^{(i)})).$ Also, our revision will include the above derivation and discussion of TBA’s gradient to clarify its relationship to other methods. We believe this greatly augments our original discussion of $\log \hat{Z}$ as a control variate (see Line 131), allowing readers to see exactly how $\log \hat{Z}$ plays a role similar to the average of rewards that GRPO computes and includes as a control variate.
>
>
> Finally, as an aside, please note that the reward functions are the same in Equations 4 and 5 – the absence of the $\phi$ subscript from the reward function in Equation 5 is a typo. We have corrected this in our revision.
>
>
> > GFlowNet is already an established method. The paper does not offer innovation on methodology beyond the existing method. However, applying it for LLM post-training is interesting, and the empirical explorations seem warranted.
>
>
> We would like to emphasize that we propose an asynchronous version of TB – Trajectory Balance with Asynchrony (TBA) – that had not previously been tested or suggested (even outside LLM post-training), which is notable given the suitability of TB’s off-policy-compatibility to the off-policy data generated in asynchronous LLM RL settings. Moreover, our implementation of TBA is efficient and effectively post-trains language models much faster than existing algorithms. Finally, we note that we go beyond prior work on GFlowNets by testing TBA’s
> - scalability to 64+ GPUs in a distributed setting,
> - compatibility with a range of buffer sampling techniques (e.g., to prioritize reward or recency),
> - sensitivity to off-policy samples, and
> - hyperparameters in various ablation studies.
>
> We will use the extra space available in the revised paper to emphasize these points, clarifying the novelty of our research.

---

> ### Comment · Reviewer_cZos · 2025-08-05
>
> I really appreciate the additional discussion, especially around the objective.
>
> Reading your equation, which I appreciate, you essentially have a transformed reward that has an importance sampling term:
>
> $\log \frac{\pi_{ref}}{\pi_{\theta}} + r - \log \hat Z(x)$
>
> This is actually quite different. GRPO's score is usually of the form:
>
> $\frac{\pi_{ref}}{\pi_{\theta}} ( r  - b(x))$
>
> In this case, $b(x)$ is the control variate.
>
> Is one way to think of $Z$ as if it's giving an exploration bonus?
>
> I do think this paper should be accepted and will state my opinion as such -- but would like to know the difference between TBA and GRPO.
>
> The scalability is not a good claim because GRPO and most RL methods already scale. TBA scales similarly to those, but perhaps you were comparing it to the original Trajectory Balancing paper.

---

> > ### Author Response · Authors · 2025-08-06
> >
> > > I do think this paper should be accepted and will state my opinion as such
> >
> > Thank you for your continued engagement and interest in supporting our work.
> >
> > > Reading your equation, which I appreciate, you essentially have a transformed reward that has an importance sampling term... Is one way to think of as if it's giving an exploration bonus?
> >
> > This is roughly correct, and we will add some additional clarifications below. Please first note that our response to Reviewer C5mN contains a helpful comparison of the policy probability ratios in TBA and Proximal RLOO (a GRPO-like off-policy method) that we will include in our revision. That discussion may add additional clarity, along with the following.
> >
> > The ratio that shows up in TBA weights rewards by placing the trained policy $\pi_{\theta}$ in the denominator: $\frac{\pi_{\text{ref}}}{\pi_{\theta}}$. Note that this is not an importance sampling ratio like the ratios in PPO/GRPO/etc. Instead, by weighting rewards by $\frac{\pi_{\text{ref}}}{\pi_{\theta}}$, TBA encourages exploration of sequences that are probable under the original reference policy $\pi_{\text{ref}}$ (e.g. the base LLM).
> > - As a side note, and as discussed with Reviewer C5mN, the control variate in TBA $\log \hat{Z}(x)$ incorporates deviations from the reference policy in its calculation $\log \hat{Z}(x) = \frac{1}{k} \sum\_{i}  \bigg( \log \pi_{\text{ref}}(y_i \mid x) - \log \pi_{\theta}(y_i \mid x) + \frac{1}{\beta} r(y_i; x) \bigg).$
> >
> > In contrast, PPO/GRPO/etc. weight rewards by placing the trained policy in the numerator rather than the denominator: $\frac{\pi_{\theta}}{\pi_{\theta'}}$. Here, $\theta'$ is the copy of the policy used to generate rollouts, and this ratio facilitates importance sampling.
> > - As another side note and consistent with your understanding, the control variate in GRPO $b(x)$ is not computed as a function of the probability ratio, unlike $\log \hat{Z}(x)$.
> >
> > In summary, TBA uses a unique probability ratio that facilitates exploration rather than importance sampling. For more information on the fundamental distinction between policy gradient approaches like GRPO and trajectory balance, our revision will refer readers to work on path consistency learning (Nachum et al., 2017): indeed, TBA’s use of a consistency objective allows it to learn from off-policy data without any additional correction mechanisms.
> >
> >  > The scalability is not a good claim because GRPO and most RL methods already scale. TBA scales similarly to those, but perhaps you were comparing it to the original Trajectory Balancing paper.
> >
> > Our revision will clarify that we are, as you suggested, demonstrating the scalability of TBA relative to prior work on TB. Additionally, we demonstrate the scalability of the components of our asynchronous training framework: e.g., we show that the TBA codebase can efficiently populate a training process’s buffer using generations from dozens of distributed GPUs, enabling faster training while also achieving enhanced exploration of model generations.

---

> > > ### Author Response · Authors · 2025-08-08
> > >
> > > Hello cZos,
> > >
> > > As the discussion period comes to a close, we would like to thank you once more for your thoughtful engagement with our submission. Your questions and suggestions have strengthened our work.
> > >
> > > We greatly appreciate your positive assessment of our paper. Since you mentioned that you “do think this paper should be accepted,” we would be grateful if you could update your official rating to reflect this recommendation.
> > >
> > > Regarding your questions about the differences between TBA and GRPO, we believe the above replies address them and will include these clarifications in the revision.
> > >
> > > Thank you again for your time and careful consideration of our work. Your feedback has been instrumental in improving the clarity and quality of our contribution.
> > >
> > > Authors of 11130

---

### Note · Authors · 2025-08-12

Thank you again, reviewers and AC. We are encouraged by the reactions to our rebuttal: **cZos** wrote “I do think this paper should be accepted and will state my opinion as such” and **ZkfV** gave an accept rating of 5.

We hope all reviewers and the AC consider recommending the paper for acceptance.

---
### Our Contribution

We introduce the first *asynchronous* Trajectory Balance (TBA) framework for LLM post-training. TBA decouples search from learning, fills a central replay buffer in a scalable manner, and trains with an off-policy TB objective using reward+recency sampling from the buffer.

Our submission showed
- **wall-clock speedups** of ~4x vs. synchronous methods like GRPO and 50x vs methods that require extra rollouts (e.g., VinePPO), all in compute-matched (4 GPU) settings;
- **state-of-the-art accuracy/win-rate** while training off-policy that matches or surpasses on-policy methods’;
- **scalability** shown up to 64 GPUs without sacrificing speedups.

### Improvements from discussion
- [**cZos**, **C5mN**, **2z92**, **ZkfV**] **Clearer theory:** We derived the relationship between TBA’s gradient and the gradient for GRPO/Proximal-RLOO, adding discussion of their advantage-like terms. We also clarified why trajectory balance optimization converges with the delayed policies that arise under asynchronous training.
- [**C5mN**] **Limited-resource regimes:** Our original submission includes compute-matched results for GSM8K and TL;DR, which use one node (4 GPUs). TBA is much faster and the most accurate in this practical small-scale setting, and our revision will emphasize this.
- [**2z92**] **Robustness to reward noise/hacking:** When optimizing imperfect proxy rewards, TBA improves the Pareto frontier (gold-standard reward vs. reference-policy perplexity) relative to widely used RLHF baselines, indicating higher robustness. Our revision will better discuss the importance of this result.
- [**C5mN**, **2z92**] **Stronger off-policy baselines:** In new results, TBA achieves the highest win rate and lowest reference model perplexity vs. Async-PPO, Proximal-RLOO, and Online-DPO (**C5mN**, **Table R2**).
- [**2z92**] **Variance & efficiency:** Doubling K (responses per prompt) modestly improves accuracy and reduces variance at no extra rollout cost.
- [**C5mN**, **2z92**] **Larger model size and context window:** Our revision will add the experiment training a 7B parameter model on MATH with TBA 70.0% and Async-GRPO 67.3% (**C5mN**, **Table R1**).

---

### Decision · Program_Chairs · 2025-09-17

**Decision:**

Accept (poster)

**Comment:**

The work makes solid technical contributions by decoupling data generation from policy updates and demonstrates effectiveness across mathematical reasoning, preference tuning, and red-teaming tasks. Two reviewers were strongly supportive: cZos explicitly stated "I do think this paper should be accepted" and raised their score to 5, while ZkfV increased to Strong Accept (6) after discussion.

Authors provided comprehensive rebuttals addressing theoretical concerns, adding 7B model experiments, and comparing against stronger off-policy baselines. The main issue was reviewer C5mN's non-participation in required discussions despite thorough responses to their concerns.

The asynchronous framework addresses real scalability challenges in LLM post-training with convincing experimental validation. This represents meaningful progress in making RL-based LLM training more efficient and practical.